# The Response of the Amazon Ecosystem to the Photosynthetically Active Radiation Fields: Integrating Impacts of Biomass Burning Aerosol and Clouds in the NASA GEOS Earth System Model

Huisheng Bian[1,2], Eunjee Lee[3,4], Randal D. Koster[4], Donifan Barahona[4], Mian Chin[2], Peter R. Colarco[2], Anton Darmenov[2], Sarith Mahanama[5,4], Michael Manyin[5,4], Peter Norris[3,4], John Shilling[6], Hongbin Yu[2], and Fanwei Zeng[5,4]

[1]Joint Center for Environmental Technology UMBC, Baltimore, MD, 21250, USA
[2]Laboratory for Atmospheres, NASA Goddard Space Flight Center, Greenbelt, MD, 20771, USA
[3]Goddard Earth Sciences Technology and Research, Universities Space Research Association, Columbia, MD 21046, USA
[4]Global Modeling and Assimilation Office, NASA Goddard Space Flight Center, Greenbelt, MD 20771, USA
[5]Science Systems and Applications, Inc., Lanham, MD 20706, USA
[6]Atmospheric Sciences & Global Change Division, Pacific Northwest National Laboratory, Richland, WA 99352, USA

*Correspondence to*: Huisheng Bian (Huisheng.Bian@nasa.gov)

**Abstract**

The Amazon experiences fires every year, and the resulting biomass burning aerosols, together with cloud particles, influence the penetration of sunlight through the atmosphere, increasing the ratio of diffuse to direct photosynthetically active radiation (PAR) reaching the vegetation canopy and thereby potentially increasing ecosystem productivity. In this study, we use the NASA Goddard Earth Observing System (GEOS) model with coupled aerosol, cloud, radiation, and ecosystem modules to investigate the impact of Amazon biomass burning aerosols on ecosystem productivity, as well as the role of the Amazon's clouds in tempering this impact. The study focuses on a seven-year period (2010-2016) during which the Amazon experienced a variety of dynamic environments (e.g., La Niña, normal years, and El Niño). The direct radiative impact of biomass burning aerosols on ecosystem productivity—called here the aerosol diffuse radiation fertilization effect —is found to increase Amazonian Gross Primary Production (GPP) by 2.6% via a 3.8% increase in diffuse PAR (DFPAR) despite a 5.4% decrease in direct PAR (DRPAR) on multiyear average during burning seasons. On a monthly basis, this increase in GPP can be as large as 9.9% (occurring in August 2010). Consequently, the net primary production (NPP) in Amazon is increased by 1.5%, or ~92 TgCyr$^{-1}$– equivalent to ~37% of the average carbon lost due to Amazon fires over the seven years considered. Clouds, however, strongly regulate the effectiveness of the aerosol diffuse radiation fertilization effect. The efficiency of this fertilization effect is the highest in cloud-free conditions and linearly decreases with increasing cloud amount until the cloud fraction reaches ~0.8, at which point the aerosol-influenced light changes from being a stimulator to an inhibitor of plant growth. Nevertheless, interannual changes in the overall strength of the aerosol diffuse radiation fertilization effect are primarily controlled by the large interannual changes in biomass burning aerosols rather than by changes in cloudiness during the studied period.

47
## 1. Introduction

The Amazon is home to more than 34 million people and hosts a large variety of plants and animals. The rainforest plays a vital role in the global climate, regulating temperatures and storing vast quantities of carbon (Laurance 1999; Nepstad et al., 2008). It is matter of intense research whether light or water is the limiting factor that controls plant growth over Amazonia. Considerable evidence demonstrates that sunlight indeed drives Amazon forest growth (Doughty et al., 2019; Huete et al., 2006; Myneni et al., 2007) although water deficit could be a limiting factor during severe droughts (Doughty et al., 2015; Feldpausch et al., 2016; Saatchi et al., 2013). Satellite observations show a clear seasonal cycle with a gradual crescendo in both leaf area and incoming surface sunlight beginning at the onset of the dry season (~August – November) (Myneni et al., 2007). Vegetation index maps also show that a majority of Amazonia is greener in the dry season than in the wet season (~mid-December – mid-May) (Huete et al., 2006). It is in the dry season, when more light reaches the canopy level, that the Amazon forest thrives.

Plant photosynthesis requires sunlight to reach the leaves of the canopy. While aerosols and clouds in the atmosphere decrease the total amount of light that reaches the canopy, they also increase scattering, thereby increasing the ratio of diffuse radiation to direct radiation. This is important because the efficiency of plant photosynthesis increases under diffuse sunlight – a phenomenon both explained theoretically (Rap et al., 2015; Roderick et al., 2001; Zhou et al., 2020) and observed in the field (Cirino et al., 2014; Doughty et al., 2010; Ezhova et al., 2018; Gu et al., 2003; Lee et al., 2018; Niyogi et al., 2004; Oliveira et al., 2007). Leaf photosynthesis increases nonlinearly with solar radiation, becoming saturated on bright days at light levels above which leaves cannot take more light (Gu et al., 2003; Mercado et al., 2009). Under clear and clean sky conditions, particularly around midday, sunlight is mainly direct, and while this allows the sunlit leaves on top to be light saturated, the shaded leaves below them receive relatively little sunlight and thus participate less in photosynthesis (Rap et al., 2015; Roderick et al., 2001). In contrast, under cloudy conditions or in the presence of aerosols, much of the midday light is diffuse, and diffuse light can penetrate deeper into the canopy and illuminate shaded leaves. Li and Yang (2015) conducted a chamber experiment to explore diffuse light on light distribution within a canopy and the resulting effects on crop photosynthesis and plant growth. They concluded that diffusion of the incident light improves spatial light distribution, lessens the variation of temporal light distribution in the canopy, and allows more light-stimulated growth of shade-tolerant potted plants.

The situation is more profound during the Amazon dry season when intensive seasonal fires release large amounts of primary aerosol particles as well as gas precursors that form secondary organic and inorganic aerosols. Using stand-alone radiation and vegetation models, Rap et al. (2015) concluded that fires over the Amazon dry season increase Amazon net primary production (NPP) by 1.4–2.8% by increasing diffuse radiation. This enhancement of Amazon basin NPP (78–156 Tg C a$^{-1}$) is equivalent to 33–65% of the annual regional carbon emissions from biomass burning and accounts for 8–16% of the observed carbon sink across mature Amazonian forests. Moreira et al. (2017) advanced this analysis by coupling an ecosystem module and aerosol model within a Eulerian transport model. Their study indicated that biomass burning aerosols lead to increases of about 27% in Amazonian Gross Primary Production (GPP)

and 10% in plant respiration as well as a decline in soil respiration of 3 %. However, their
approach assumes cloud-free conditions through their use of a diffuse irradiance
parameterization based on the multiwavelength aerosol optical depth (AOD) measurement.
Malavelle et al. (2019) explored the overall net impact of biomass burning aerosol on the
Amazon ecosystem using an Earth System Model (ESM) (HadGEM2-ES). They estimated NPP
to increase by +80 to +105 TgC yr$^{-1}$, or 1.9% to 2.7%, ascribing this net change to an increase in
diffuse light, a reduction in the total amount of radiation, and feedback from climate adjustments
in response to the aerosol forcing. Their study takes into account the dynamic feedback of short
lifetime cloud fields. However, the authors have not explicitly quantified the impact of Amazon
background clouds and their interannual changes in tempering the aerosol diffuse radiation
fertilization effect (DRFE).
When clouds and aerosol co-exist, the impact from clouds on the ecosystem typically dominates
because clouds are optically thicker. The surface sunlight for cloudy versus cloud-free conditions
can differ greatly even if the AOD is the same. (Note that, unless specified otherwise, solar
radiation in this study refers to the wavelength range of 400-700 nm, i.e., photosynthetically
active radiation, or PAR). Measurements indicate that the desirable range of clearness index (CI)
-- the ratio of total (i.e., direct plus diffuse) light at surface to the total incoming light at top of
atmosphere -- is around 0.4-0.7 for some forest ecosystems and above 0.3 for peatland (Butt et
al., 2010, Letts and Lafleur, 2005). Quite often a low CI occurs during a cloudy day, but on
occasion it might result from the presence of a very thick aerosol layer. As suggested above, if
CI is high, the diffuse fraction of the total solar radiation is low, and the overall productivity of
the canopy is reduced. For example, Cirino et al. (2014) found that the net ecosystem exchange
(NEE) of $CO_2$ is increased by 29% and 20% in two Amazon stations, the Jaru Biological Reserve
(RBJ) and the Cuieiras Biological Reserve at the K34 Large-Scale Biosphere-Atmosphere
Experiment in Amazonia (LBA) tower, respectively, when AOD is 0.1-1.5 at 550nm under clear
conditions. Higher AOD (> 3) leads to a strong reduction in photosynthesis (via reducing PAR)
up to the point where NEE approaches zero. Oliveira et al. (2007) found that Amazon forest
productivity was enhanced under moderately thick smoke loading because of an increase of
diffuse solar radiation, but large aerosol loading (i.e., AOD > 2.7) results in lower net
productivity of the Amazon forest.
Despite its name, the Amazon's "dry season" (June-November) still features significant
cloudiness, and the interannual variations in the clouds can be large. Furthermore, rain does fall
during the dry season – close to 40% of the total annual precipitation falls therein (Li et al.,
2006). Clouds in the dry season are mostly formed by small-scale processes that influence the
weather (see an example of a uniform layer of "popcorn" clouds observed by Moderate
Resolution Imaging Spectroradiometer (MODIS) on 08/19/2009 in
http://earthobservatory.nasa.gov/IOTD/view.php?id=39936). It is during this period, when
sunlight (particularly diffuse light) shines on the trees due to reduced rain (and fewer clouds)
relative to the wet season, that the forest grows the most. Consideration of the joint effects of
clouds and biomass burning aerosols on diffuse and direct PAR during the dry season is thus
particularly important.
This study has two objectives. First, we investigate how Amazon biomass burning aerosols
(BBaer) affect the land productivity (i.e., GPP and NPP) via their impact on direct and diffuse
PAR (DRPAR and DFPAR). Second, we investigate the sensitivity of the BBaer DRFE to the
presence of the Amazon dry season cloud fields within the range indicated by the interannual
variation of the clouds. We use in our analysis a version of the NASA GEOS ESM that includes
coupling between aerosol, cloud, radiation, and ecosystem processes. To our knowledge, only
one other study has used an ESM to investigate such fire impacts across Amazonia (Malavelle et
al., 2019), and as noted above, that study did not address the ability of Amazon clouds to temper
the BBaer impacts. Accordingly, our study is the first ESM-based study to investigate the BBaer
DRFE within a range of interannual Amazon cloud levels. Together our objectives provide a full
and comprehensive study of BBaer DRFE in a context of potential Amazon dry season
atmospheric conditions.
It is necessary to point out, however, that our study focuses only on the impact of Amazon
biomass burning aerosol. We do not consider the radiative impacts of other potentially important
aerosols. These other aerosol types have been examined in various observational studies (e.g.,
Cirino et al., 2014; Ezhova et al., 2018; Hemes et al., 2020; Wang et al., 2018, Yan et al., 2014)
and model investigations that focus, for example, on anthropogenic aerosol (Keppel et al., 2016);
O'Sullivan et al., 2016), dust (Xi et al., 2012), biogenic aerosol (Rap et al., 2018; Sporre et al.,
2019), volcanic aerosol (Gu et al., 2003), and the general aerosol field (Feng et al., 2019).
The paper is organized as follows. Section 2 describes the NASA GEOS ESM and its relevant
modules (section 2.1), the observational data used for model evaluation and explanation (section
2.2), and the experimental setup (section 2.3). Section 3 provides an evaluation of the model
(section 3.1), basic theory regarding the impact of aerosol and cloud on the surface downward
radiation (section 3.2), results regarding the simulated ecosystem response to BBaer-induced
radiation changes (section 3.3), and the impacts of Amazon background clouds on this response
(section 3.4). A final summary is provided in section 4.
**2. Model description, data application, and experiment setup**
**2.1 Model description**
The GEOS modeling system connects state-of-the-art models of the various components of the
Earth's climate system together using the Earth System Modeling Framework (ESMF) (Molod et
al., 2015; 2012; Rienecker et al., 2011; https://gmao.gsfc.nasa.gov/). We discuss here the
components of the system that are particularly relevant to our study, including aerosol, cloud,
microphysics, radiative transfer, and land ecosystem modules.
GEOS Goddard Chemistry Aerosol Radiation and Transport (GOCART) simulates a number of
major atmospheric aerosol species and precursor gases from natural and anthropogenic sources,
including sulfate, nitrate, ammonium, black carbon (BC), organic aerosol (OA, including
primary and secondary OA), dust, sea salt, dimethyl sulfide (DMS), $SO_2$, and $NH_3$ (Bian et al.,
2010, 2013, 2017, 2019; Chin et al., 2009, 2014; Colarco et al., 2010, 2017; Murphy et al., 2019;
Randles et al., 2013). Monthly emissions from shipping, aircraft, and other anthropogenic
sources are obtained from the recent CMIP6 CEDS emission inventory. Daily biomass burning
emissions are provided by GFED4s
(https://daac.ornl.gov/VEGETATION/guides/fire_emissions_v4.html). Estimates of degassing
and eruptive volcanic emissions are derived from Ozone Monitoring Instrument (OMI) satellite
(Carn et al., 2017). Emissions of dust, sea salt, and DMS are dynamically calculated online as a
function of the model-simulated near-surface winds and other surface properties. A more recent
development of GOCART relevant to this study involves the modification of the absorbing
properties of "brown carbon" from biomass burning organic aerosols (Colarco et al., 2017) and
the inclusion of secondary organic aerosol (SOA) produced via chemical reactions of volatile
organic compounds (VOCs) emitted from anthropogenic and biomass burning sources, following
the approach developed by Hodzic and Jimenez (2011) and Kim et al. (2015). In addition, the
SOA from biogenic sources has been updated with its precursor gases of isoprene and
monoterpene emissions calculated online as a function of light and temperature using the Model
of Emissions of Gases and Aerosols from Nature (MEGAN) version 2.1 (Guenther et al., 2012),
assuming SOA yield of 3% from isoprene and 5% from monoterpene oxidations (Kim et al.,
196 2015).
The GEOS two-moment cloud microphysics module is used in this study. The module includes
the implementation of a comprehensive stratiform microphysics module, a new cloud coverage
scheme that allows ice supersaturation, and a new microphysics module embedded within the
moist convection parameterization (Barahona et al., 2014). At present, aerosol number
concentrations are derived from the GEOS/GOCART-calculated aerosol mass mixing ratio and
prescribed size distributions and mixing state, which are then used for cloud condensation nuclei
(CCN) activation (following the approach of Abdul-Razzak and Ghan, 2000) and ice nucleation
(following the approach of Barahona and Nenes, 2009) processes. Aerosol-cloud interactions are
thus accounted for in our simulation. The model calculates various cloud properties, including
cloud fraction, cloud droplet and ice crystal number concentrations and effective radii, and cloud
liquid and ice water paths. These fields have been evaluated against satellite observations and
field measurements; the model shows a realistic simulation of cloud characteristics despite a few
remaining deficiencies (Barahona et al., 2014, Breen et al., 2020).
The current default GEOS solar radiation transfer module is the shortwave rapid radiation
transfer model for GCMs (RRTMG_SW), a correlated k-distribution model (Iacono et al., 2008).
This GCM version utilizes a reduced complement of 112 g-points, which is half of the 224 g-
points used in the standard RRTMG_SW, and a two-stream method for radiative transfer. Total
fluxes are accurate to within 1-2 W/m$^2$ relative to the standard RRTMG_SW (using DISORT)
with aerosols in clear sky and within 6 W/m$^2$ in overcast sky.  RRTMG_SW with DISORT is
itself accurate to within 2 W/m$^2$ of the data-validated multiple scattering model, CHARTS.
RRTMG_SW specifically calculates the direct and diffuse components of PAR (400-700 nm)
separately. The GEOS atmospheric radiative transfer calculation is designed in a way that allows
users to examine the impact of various combinations of atmospheric aerosol and cloud fields on
radiation. In addition to the standard calculation of solar radiation for ambient atmospheric
conditions, diagnostic calculations can be carried out by repeating the calculation of the radiation
transfer scheme with different combinations of atmospheric conditions: clean air (no aerosols),
clear air (no clouds), and clean plus clear air. Using this architecture, for this study we modify
the radiation scheme to allow the additional diagnosis of radiation fields under conditions of zero
BBaer but retained non-BBaer and ambient clouds.
The catchment land surface model (LSM) with carbon and nitrogen physics (Catchment-CN) in
GEOS is in essence a merger of the C-N physics within the NCAR–DOE Community Land
Model (CLM) (Oleson et al. 2010, 2013; Lawrence et al., 2019) version 4.0 and the energy and
water balance calculations of the NASA GMAO catchment LSM (Koster et al. 2000). The
original NASA catchment LSM used a prescribed representation of phenology (leaf area index,
or LAI, and greenness fraction) to compute the canopy conductance, the parameter describing
the ease with which the plants transpire water. The light interception by vegetation in the GEOS
Catchment-CN utilizes the same parameterization as that in CLM4. The photosynthesis and
transpiration depend non-linearly on solar radiation. The canopy is assumed to consist of sunlit
leaves and shaded leaves, and the DRPAR and DFPAR absorbed by the vegetation is
apportioned to the sunlit and shaded leaves as described by Thornton and Zimmermann (2007).
The prognostic carbon storages underlying the phenological variables are computed as a matter
of course along with values of canopy conductance that reflect an explicit treatment of
photosynthesis physics. These canopy conductances, along with the LAIs diagnosed from the
new carbon prognostic variables, are fed into the energy and water balance calculations in the
original catchment LSM. The output fluxes from the merged system include carbon fluxes in
addition to traditional fluxes of heat and moisture. The merger of the two models allows
Catchment-CN to follow 19 distinct vegetation types. Koster and Walker (2015) have used
Catchment-CN within an atmospheric global circulation model (AGCM) framework to
investigate interactive feedback among vegetation phenology, soil moisture, and temperature. In
this study, the modeled atmospheric $CO_2$ from the AGCM is used to drive the carbon, water, and
energy dynamics in the Catchment-CN model.
In addition to the GEOS ESM, we use a photolysis scheme, FastJX, in its stand-alone mode to
explore how incoming solar radiation penetrates the atmosphere in the presence of aerosols and
clouds in order to enhance our basic understanding of the role of atmospheric particles on
radiation. FastJX is based on the original Fast-J scheme, which was developed for tropospheric
photochemistry with interactive consideration of aerosol and cloud impacts at 291–850 nm (Wild
et al., 2000), and Fast-J2, which extended the scheme into the deep UV spectrum range of 177-
291 nm (Bian and Prather, 2002).
**2.2 Observational data**
We mostly rely on the GoAmazon ("Green Ocean Amazon") field campaign
(http://campaign.arm.gov/goamazon2014/) for in-situ aerosol observations to assess the model-
simulated OA concentrations. GoAmazon is an integrated field campaign conducted in the
central Amazon Basin (Martin et al., 2016). Specifically, the following datasets are used: a) the
surface OA concentration measured in 2014 by the Aerosol Chemical Speciation Monitor
(ACSM) operated by the Department of Energy's (DOE) Atmospheric Radiation Measurement
(ARM) Mobile Facility located 70 km downwind of Manaus, Brazil (Ng et al., 2011), b) the
surface CO volume mixing ratio in 2014 at Manaus measured by Los Gatos Research (LGR)
$N_2O$/CO Analyzer that uses LGR's patented Off-axis Integrated Cavity Output Spectroscopy
(ICOS) technology, and c) the vertical profile of OA concentration measured by a time-of-Flight
Aerosol Mass Spectrometer (ToF-AMS) instrument on the ARM Aerial Facility Gulfstream-1
(G-1) aircraft during the dry season of 2014 (Sept 06-Oct 04, 2014) (Shilling et al., 2018). The
G-1 aircraft was based out of the Manaus International airport and flew patterns designed to
intersect the Manaus urban plume at increasing downwind distance from the city (e.g., 59-61°W
and 4-2.5°S). In addition, we evaluate the model with AOD and single scattering albedo (SSA)
measurements taken at a central Amazon station (Alta Floresta) in the ground-based Aerosol
Robotic Network (AERONET) sun photometer network (http://aeronet.gsfc.nasa.gov). We also
use MODIS collection 6.1 level-3 AOD product
(http://modis.gsfc.nasa.gov/data/dataprod/index.php), which is characterized by observations
with large spatial coverage.
MODIS cloud products (https://modis-atmosphere.gsfc.nasa.gov/data/dataprod/), specifically
total cloud fraction and cloud optical depth in liquid and ice particles, are used to evaluate the
model cloud simulation. We use the cloud data from MODIS collection 6.1 MYD08_D3, a level-
3 1°×1° global gridded monthly joint product derived from the MODIS level-2 pixel level
products. MODIS level 2 cloud fraction is produced by the infrared retrieval methods during
both day and night at a 5×5 1-km-pixel resolution. Level 2 cloud optical thickness used in this
study is derived using the MODIS visible and near-infrared channel radiances from the Aqua
platform.
The satellite-derived Clouds and the Earth's Radiant Energy System product CERES-EBAF is
used to evaluate the GEOS simulation of radiation fields. CERES-EBAF retrieves surface
downward shortwave radiation ($R_{SFC}$) using cloud information from more recent satellite data
(MODIS, CERES, CloudSat and CALIPSO) and aerosol fields from AERONET/MODIS
validation-based estimates (Kato et al., 2013). This global product is provided at a 1°×1°
horizontal resolution and covers the years 2000-2015 for both all- and clear-sky conditions. The
multiyear $R_{SFC}$ products provide both a spatial and temporal view of radiation over Amazonia.
Two observation-based GPP products (FluxCom and FluxSat) are used to evaluate ecosystem
productivity in the GEOS simulations. The FluxCom GPP product provides globally distributed
eddy-covariance-based estimates of carbon fluxes between the biosphere and the atmosphere
through upscaling using machine learning methods (Jung et al., 2020). FluxSat GPP is estimated
with models that use satellite data (e.g., MODIS reflectances and solar-induced fluorescence
(SIF)) within a simplified light-use efficiency framework (Joiner et al., 2018). We use monthly
GPP for August through October of 2010-2015 in this study.
**2.3 Experiment setup**
All experiments were run with the coupled atmosphere and land components of the NASA
GEOS ESM system discussed above. The sea surface temperature (SST) for the atmospheric
dynamic circulation is provided by the GEOS Atmospheric Data Assimilation System (ADAS)
that incorporates satellite and in situ SST observations and assimilates Advanced Very High
Resolution Radiometer (AVHRR) brightness temperatures. The experiments were run in replay
mode, which means that the model dynamical variables (winds, pressure, temperature, and
humidity) were set, every 6 hours, to the values archived by the Modern-Era Retrospective
Analysis for Research and Applications version 2 (MERRA-2) meteorological reanalysis (Gelaro
et al. 2017); a 6-hourly forecast provided the dynamical and physical fields between the 6-hour
resets. In effect, the replay approach forces the atmospheric "weather" simulated in the model to
agree with the reanalysis. This nudging of the GEOS dynamic fields toward the MERRA2
reanalysis ensures that the atmospheric conditions of our four simulations (see below) remain
close to each other, allowing a more focused study of radiative impact on ecosystem. All
designed experiments were run over 2010-2016, a period that includes La Niña (2010-2011), El
Niño (2015-2016), and neutral years as indicated by the Oceanic Niño Index (ONI,
https://origin.cpc.ncep.noaa.gov/) (Figure S1). Information regarding long-term BB OA
emissions (i.e., 1997-2016) and long-term MERRA2 cloud fraction anomalies (i.e., 1995-2018)
is shown in Figure S2. The selected period of 2010-2016 represents well the long-term period in
terms of the variation of BB emissions and cloud coverage.

Our experimental design makes extensive use of GEOS's highly flexible configuration. First, the
GEOS GOCART module includes a tagged aerosol mechanism. Each specific aerosol
component in GOCART is simulated independently from the others, and the contribution of each
emission type to the total aerosol mass is also not interfered by that of other emission types.
Thus, additional aerosol tracers can easily be "tagged" according to emission source types. This
makes it possible for GOCART to calculate and transfer two sets of aerosol fields (e.g., one with
and one without a biomass burning source) to the radiation module. Second, the radiation module
can in turn calculate a set of atmospheric radiation fields corresponding to each set of aerosol
fields, and it can then disseminate both sets of radiation fields to the various components of
interest (i.e., cloud module, land ecosystem module, etc.) according to the needs of our
experiments (see below).

Table 1 provides a brief summary of the experiments performed for this study. First, we designed
a pair of experiments (allaer and nobbaer, hereafter referred to as "pair1") to explore the BBaer
DRFE on the land productivity via PAR (objective 1). The allaer and nobbaer experiments are
designed to simulate the same atmospheric dynamics but send different PAR fluxes into the
Catchment-CN model. Specifically, both the allaer and nobbaer experiments used all
atmospheric aerosols including real-time biomass burning emissions over 2010-2016 to calculate
a set of radiation fields ($R^1$) to drive atmospheric circulation; however, with the help of GEOS's
flexible configuration, the nobbaer experiment also calculated a second set of radiation fields
($R^2$) that used non-BB aerosols only. $R^1$ was sent to Catchment-CN in the allaer experiment
whereas $R^2$ was sent to Catchment_CN in the nobbaer experiment. In this way, the only
difference between the allaer and nobbaer experiments was the PAR fluxes used to drive the
ecosystem model – only the PAR fluxes used in allaer reflected the presence of biomass burning
aerosols. The atmospheric meteorological fields in the two experiments, including clouds, skin
temperature, and soil moisture, show only minor differences stemming from land feedback
(Figure S3-4, Table 3, Table S1e and Table S2e). A negligible impact on cloud fields has also
been reported in Pedruzo-Bagazgoitia et al. (2017).

Table 1. Designed experiments (2010-2016) with their perturbation on aerosol fields and
subsequent impact on radiation and ecosystem

| Exp Name | | Aerosol | R in RRTMG | R driving Atmosphere | R driving Catchment-CN | Purpose |
|---|---|---|---|---|---|---|
| Pair 1 | allaer | Standard all, w/ Realtime AERbb emission | $R^1_{top}$, $R^1_{dir}$, $R^1_{diff}$ (all aerosol) | $R^1_{top}$, $R^1_{dir}$, $R^1_{diff}$ | $R^1_{dir}$, $R^1_{diff}$ | Check atmospheric BB aerosol impact on plants via radiation fields during 2010-2016 |
| | nobbaer | | $R^1_{top}$, $R^1_{dir}$, $R^1_{diff}$ (all aerosol) $R^2_{top}$, $R^2_{dir}$, $R^2_{diff}$ (all non-bb aerosol) | $R^1_{top}$, $R^1_{dir}$, $R^1_{diff}$ | $R^2_{dir}$, $R^2_{diff}$ | |
| Pair 2 | callaer | Standard all, w/ AERbb emission fixed at 2010 | $R^1_{top}$, $R^1_{dir}$, $R^1_{diff}$ (all aerosol) | $R^1_{top}$, $R^1_{dir}$, $R^1_{diff}$ | $R^1_{dir}$, $R^1_{diff}$ | Check how clouds adjust the above impact |
| | cnobbaer | | $R^1_{top}$, $R^1_{dir}$, $R^1_{diff}$ (all aerosol) $R^2_{top}$, $R^2_{dir}$, $R^2_{diff}$ (all non-bb aerosol) | $R^1_{top}$, $R^1_{dir}$, $R^1_{diff}$ | $R^2_{dir}$, $R^2_{diff}$ | |


We also designed a pair of experiments (callaer and cnobbaer, hereafter referred to as "pair2") to
address the sensitivity of the BBaer DRFE to the presence of the Amazon dry season cloud fields
(objective 2). The pair2 experiments are similar to those in pair1 except that the particular BB
emissions of year 2010 were repeated during all seven years. Applying a fixed aerosol emission
allows us to attribute the interannual variation of the ecosystem solely to the influence of
interannual variations in atmospheric meteorological fields, including clouds. In addition,
combining the pair1 and pair2 experiments provides two biomass burning aerosol emissions for
each year except 2010, which allows us to compare the impacts of different emissions under
similar meteorological environments (Figure S3-4, Table 3, Table S1e and Table S2e). Please
note that the experiments in this study were intentionally designed to allow the aerosols to affect
the vegetation only through their impact on the direct and diffuse radiation that enters ecosystem
and not, for example, through their other potential impacts on the environment. Future study may
focus on these other impacts. Given that the experiment period covers strong La Niña and El
Niño years, we can examine BBaer impacts on ecosystem productivity under the full range of
Amazon background cloud fields.
**3.  Results and Discussion**
**3.1 Evaluation of GEOS simulations of aerosol, cloud, radiation, and ecosystem**
**response**
The NASA GEOS ESM model, including its aerosol, cloud, radiation, and ecosystem modules as
used in the baseline simulation (i.e., experiment allaer), has been evaluated extensively and
utilized in a number of scientific studies. However, very few of the past studies with GEOS was
concentrated on detailed model evaluation over South America. We provide such an evaluation
here.
The simulated tracer fields are compared with measurements over the Amazon in Figures 1 and
2. Figure 1 shows results for surface OA concentration, surface CO concentration, and the OA
concentration vertical profile. We focus primarily on the OA evaluation since it is the major
component of biomass burning aerosols. Figure 1a shows the comparison of surface daily OA
concentration between the model simulation and the GoAmazon measurements at Manaus,
Brazil, in 2014 (The location is indicated in Figure 2c with an open-diamond). The simulated OA
broadly captures the seasonal trend in OA concentrations measured at Manaus, but it is lower
than observed OA values by ~24% during Sept-Oct and ~ 30% annually. For the period of
interest, the model simulates a large fire signal in August that is not seen in the measurements.
However, this strong August biomass burning signal does show up in the CO measurements
(Figure 1b), which should also be from biomass burning. The reasons for such discrepancy from
observations are not clear.

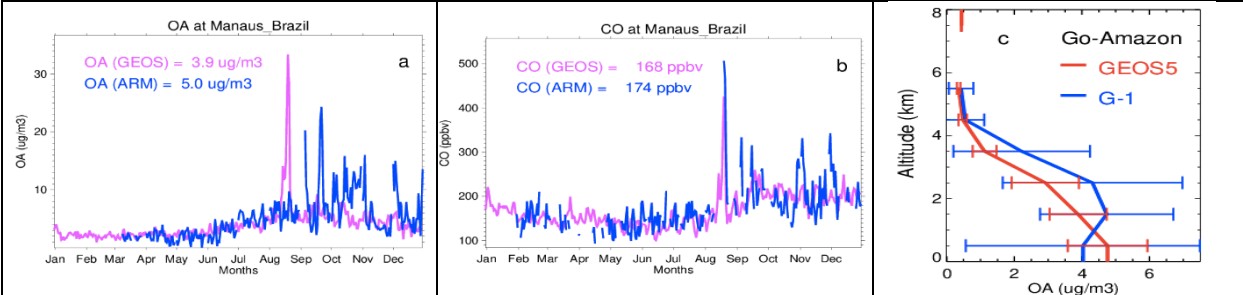

Figure 1. (a) Comparison of the ACMS measured organic aerosol (OA) daily surface mass concentration at the GoAmazon DOE ARM facility in Manaus, Brazil (location marked in Figure 2c as an open-diamond) in 2014 with GEOS simulated values. (b) Similar to (a) but for carbon monoxide (CO) volume mixing ratio. (c) GoAmazon G-1 aircraft measurement of vertical OA mass concentration during Sept 6 -Oct 4, 2014 in the vicinity of Manaus, compared to GEOS simulations. The error bars on 1c indicate one standard deviation of the data within each 1km vertical layer.


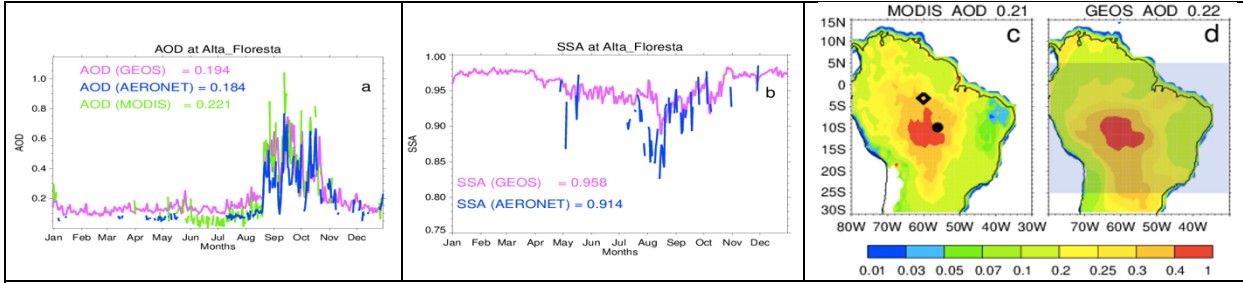

Figure 2. (a) Comparison of GEOS simulated AOD at 550nm with AERONET and MODIS daily measurements at the Alta-Floresta AERONET site for 2014. (b) similar comparison for aerosol single scattering albedo at 440nm during 2014. (c) Mean MODIS collection 6.1 AOD at 550nm during the period Aug-Oct, 2014. (d) GEOS simulated AOD at 550nm for the same period as in (c) with daily model data sampled following MODIS measurements. Note that the mean AOD values shown for (c) and (d) are averaged over the Amazon region (i.e. the shaded land area within 80°W-30°W, 25°S-5°N shown in 2d). Station locations of Alta-Floresta (filled-circle) and Manaus (open diamond) are marked in (c).


When compared with aircraft G-1 measurements over a ~2°×2° region around the center of
Manaus during the biomass burning season (Sept. 6 – Oct. 4, 2014) (Figure 1c), the simulated
vertical OA concentrations underestimate the measurements above 1 km altitude but
overestimate them under it, although they overlap within their standard deviations for all
altitudes. Here the model data have been sampled spatially and temporally along the G-1 flight
paths. This surface OA overestimation by the model seems to contradict the model's
underestimation seen in Figure 1a, indicating that capturing aerosols at the right times and
locations is a challenge.

Figure 2 shows the AOD (550nm) and SSA (440nm) comparison at the AERONET station of
Alta-Floresta, which is located close to the area of the most intensive Amazon fires (location is
marked in Figure 2c as a filled-in circle). The model-simulated, AERONET-measured, and
MODIS-retrieved AOD at this site agree within 20% (Figure 2a), all showing a peak of AOD
during the biomass burning season. SSA during the burning season generally ranges between
0.85 – 0.95 (Figure 2b). The model agrees with the measurements with accurate better than 5%
except during the first half of August, when the model aerosols are too scattering. However, it is
puzzling to observe the extremely low measured SSA in the beginning of August given that the
AOD is still low then, as shown in Figure 2a. It could be the quality of AERONET SSA is not
"reliable" at low AOD (Chin et al., 2009). Because of the low sensitivity to the absorption when
aerosol loading is low, SSA is retrieved with sufficiently high accuracy only when the
total AOD at 440 nm is equal or higher than 0.4 and solar zenith angle is 50 degree or higher
(Dubovik et al., 2000, 2002). Regionally over the Amazon region, defined throughout the study
as the land area within 80°W-30°W, 25°S-5°N (shaded land area in Figure 2d), the model-
simulated AOD (0.22 in Figure 2d) during the biomass burning season generally agrees with
MODIS satellite retrievals (0.21 in Figure 2c). A simulated high bias is seen over the east
Amazon; however, though this region is in our area of interest, the bias should have only a minor
impact on our study given that the area is relatively bare, with little vegetation coverage.

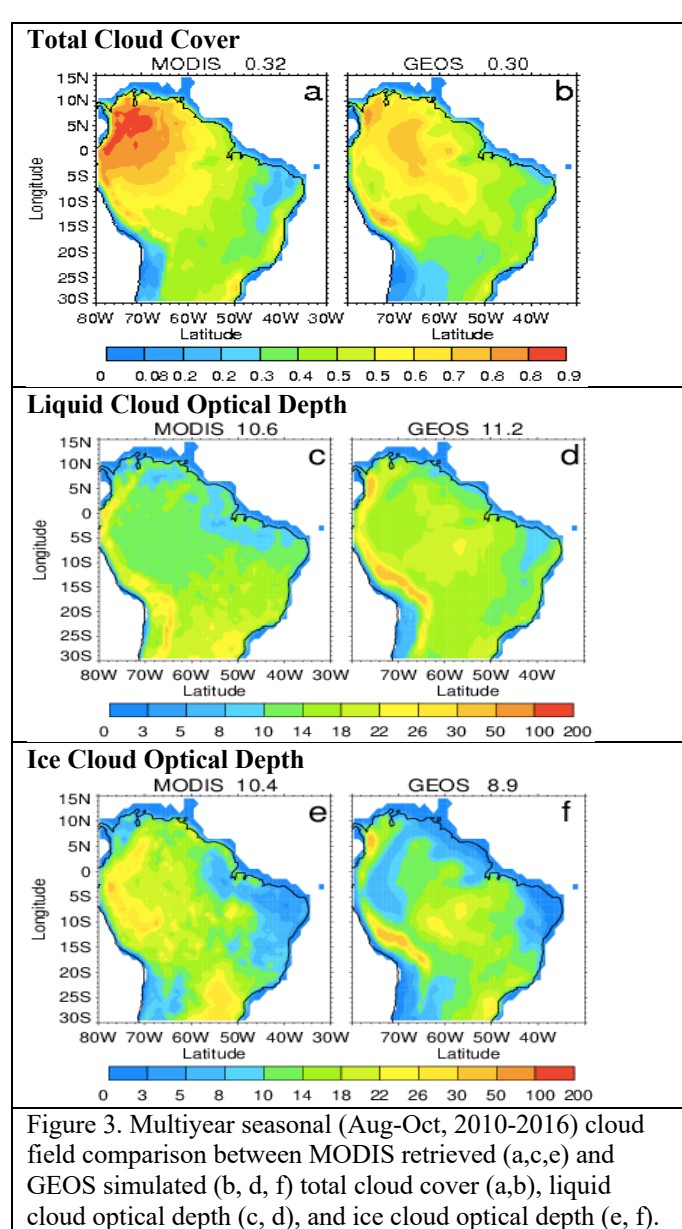

Figure 3. Multiyear seasonal (Aug-Oct, 2010-2016) cloud field comparison between MODIS retrieved (a,c,e) and GEOS simulated (b, d, f) total cloud cover (a,b), liquid cloud optical depth (c, d), and ice cloud optical depth (e, f).

The accurate simulation of cloud fields is also important for our study. In Figure 3 we evaluate
the GEOS-simulated cloud cover fraction and cloud optical depth with MODIS satellite
products. Here the GEOS data have been sampled with MODIS overpass time and location.
GEOS generally captures the magnitude and main features of the cloud fields observed in
MODIS, though with some differences; the model overestimates the cloud quantities over the
central Amazon and underestimates them in northwest South America. The overall difference
over the Amazon region between simulated and MODIS-based estimates is less than 7% for
cloud cover fraction, 10% for liquid water cloud optical depth, and 15% for ice cloud optical
depth. The seasonality of these cloud quantities is shown in Figure S5a-c to further evaluate the
model performance. The model has a better cloud simulation during the period of Aug-Oct,
which is the focus period of this study since Amazon fires occur periodically every year in this
season.
Figure 4 shows a comparison between the simulated downward shortwave radiation at the
surface and CERES-EBAF measurements averaged over Aug-Oct., 2010-2016 for both clear-sky
and all-sky conditions. The comparison of the time series of monthly mean shortwave radiation
during 2010-2016 over the Amazon region is shown in Figure S6. GEOS captures the observed
spatial patterns with ~4% high bias for both clear and all sky conditions over the Amazon region.

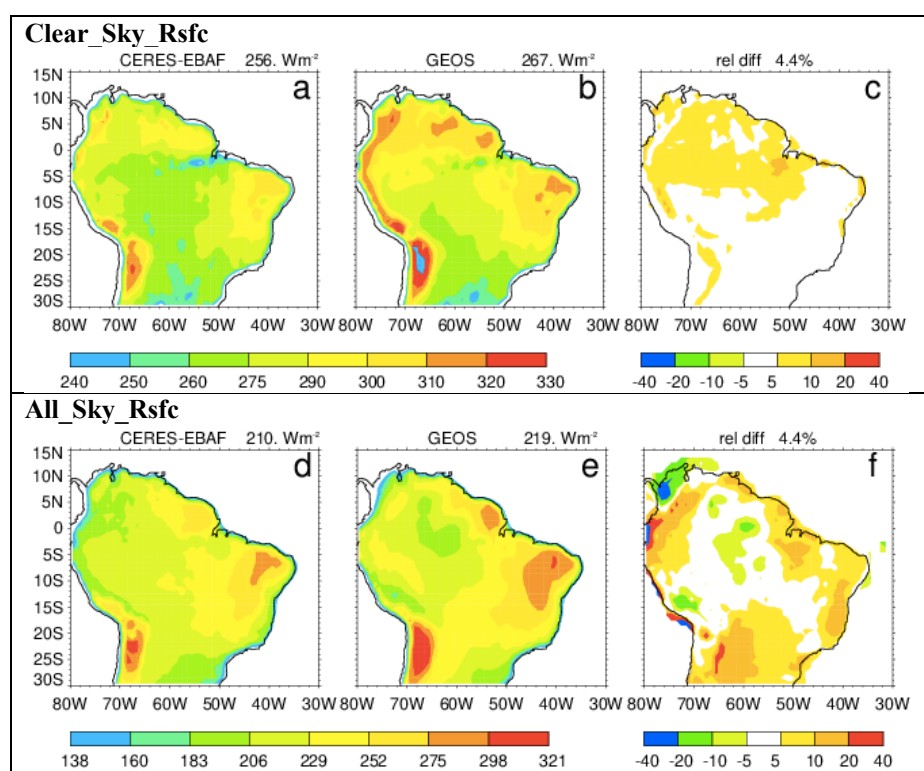

Figure 4. Comparison of surface downward shortwave radiation Rsfc (Wm$^{-2}$) between CERES-EBAF measurement and GEOS simulation averaged over Aug-Oct., 2010-2016 for clear-sky (upper panel, a, b) and all-sky (bottom panel, d, e) conditions. The right column (c, f) shows the relative difference between GEOS and CERES-EBAF.

Following the evaluation approach in Malavelle et al. (2019), we evaluate our model's ability to
simulate GPP on the global scale against FluxCom and FluxSat. As mentioned in section 2.2,
FluxCom GPP is derived from surface measurements of carbon fluxes whereas FluxSat GPP is
derived from satellite data. The comparison of global distribution of multiyear average GPP
(Figure 5) and zonal mean multiyear average GPP (Figure 6) show that GEOS captures the GPP
global distribution seen in the observations, with a GPP peak in tropics. The model does show a
second peak in middle latitudes of the Southern Hemisphere but misses the observed peak in the
Northern Hemisphere subtropics.

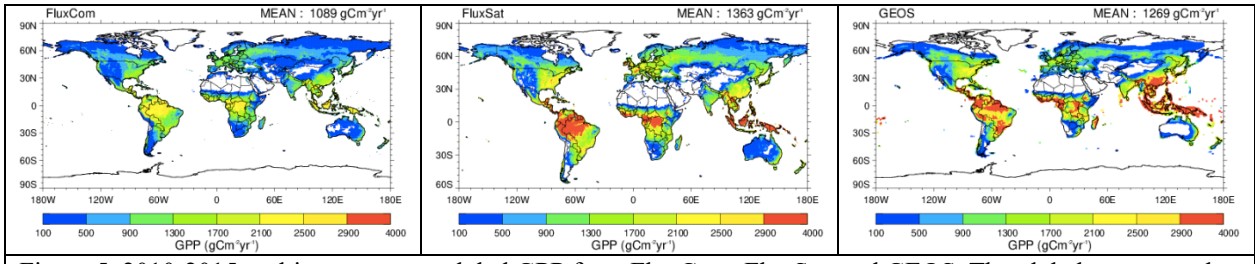

Figure 5. 2010-2015 multiyear average global GPP from FluxCom, FluxSat, and GEOS. The global average value is shown in the top.


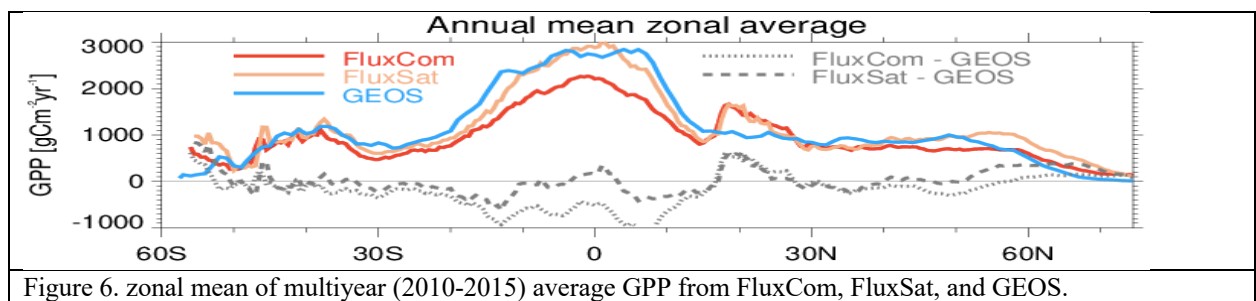

Figure 6. zonal mean of multiyear (2010-2015) average GPP from FluxCom, FluxSat, and GEOS.


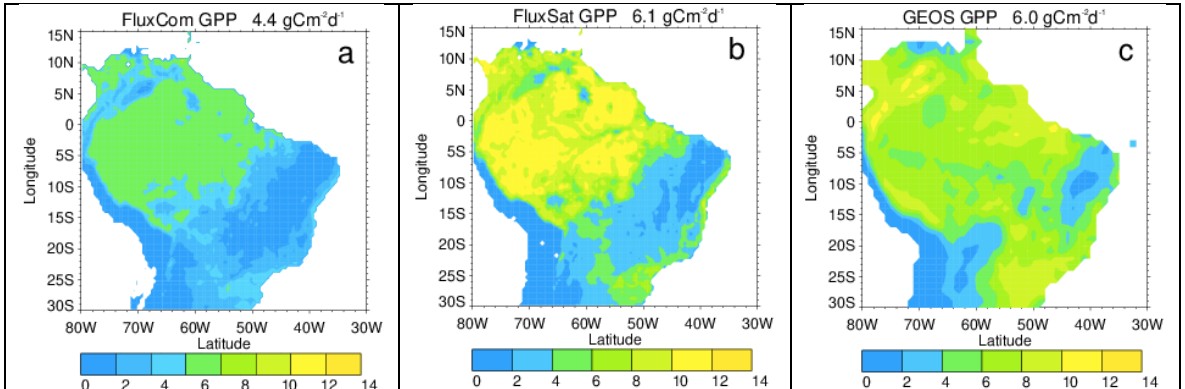

Figure 7. The multi-year (2010 – 2015) August – October mean Amazon GPP from (a) FluxCom (Jung et al., 2020), (b) FluxSat ( Joiner et al., 2018) and (c) the GEOS ecosystem simulation with unit of gC m$^{-2}$ day$^{-1}$. The Amazon regional average value is shown in the top.

Figure 7 shows GPP averaged over August to October of 2010-2015 from the two observations-
based products and the GEOS simulation. The overall spatial distributions of GEOS GPP (Figure
7c) over South America show similar spatial pattern to both of the observations-based datasets
(Figures 7a and 7b) with higher values over the eastern part of the domain but lying between the
two datasets in other areas.  Over the studied period and the Amazon region, the GEOS GPP is
comparable to the FluxSat GPP and is about 35% higher than the FluxCom GPP.
The seasonality of GPP over the Amazon region from FluxCOM, FluxSat and GEOS during
2010-2015 is shown in Figure S7, and the corresponding time series of monthly means is shown
in Figure S8. During all four seasons, regional FluxCom GPP is the lowest and FluxSat GPP is
the highest. All datasets show higher GPP during Nov-Apr than during May-Oct. GEOS
multiyear annual average GPP is close to that of FluxSat but is higher than that of FluxCom.
Although there are few of observation sites available in FLUXNET 2015 Tier 1
(https://fluxnet.org/data/fluxnet2015-dataset/), Joiner et al. (2018) evaluated FluxSat GPP
performance around Amazonia using the flux tower measurements, which showed that the high
GPP values produced by FluxSat were supported by the flux tower values (Joiner et al., 2018).

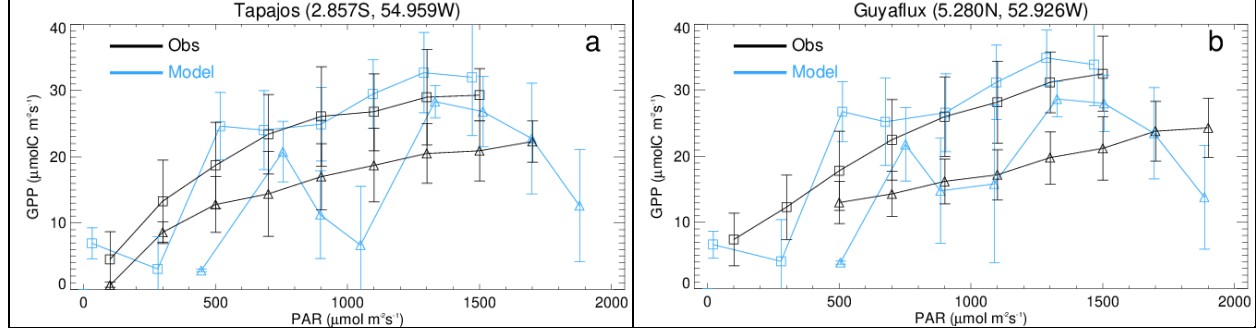

Figure 8. Observed (black) and GEOS modeled (blue) response of GPP to direct (triangles) and diffuse (squares) photosynthetically active radiation (PAR) averaged over bins of 200 μmol quanta m² s⁻¹ at (a) Tapajos and (b) Guyaflux. Error bars show 1 standard deviation of all values within a bin. The observation data, representing the period 2002-2005 for Tapajos and 2006-2007 for Guyaflux, are taken from Figure 2 of Rap et al. (2015), whereas the model period is 2010-2016 for both sites.

Although the evaluations of global and regional multiyear average GPP conducted above
(Figures 5-7) are needed for the examination of the model's fundamental mechanisms including
photosynthesis, a more direct evaluation to address the model's accuracy in simulating observed
GPP response to changes in diffuse and direct surface radiation is shown in Figure 8. Following
the evaluation approach of Rap et al., (2015), we compared the GPP response to direct and
diffuse light at two Amazon sites, Tapajos and Guyaflux. The figure clearly demonstrates that in
the model, as in observations, diffuse light is more efficient in stimulating GPP.
**3.2 Principle of aerosol and cloud impact on surface downward radiation**
Radiative responses to aerosols and cloud fields are nonlinear. To better explain the phenomenon
examined here – that plant growth increases at low-to-intermediate AOD but decreases at high
AOD – we ran the column version of a radiation model, fast-JX (Wild et al., 2000; Bian and
Prather, 2002). Fast-JX solves the 8-stream multiple scattering in atmospheric solar radiation
transfer for direct and diffuse beams, using the exact scattering phase function and optical depths
of atmospheric molecules, aerosols, and clouds, and provides photolytic intensities accurate
typically to better than 3%, with worst case errors of no more 10% over a wide range of
atmospheric conditions (Wild et al., 2000). No special approximations are needed to treat
strongly forward-peaked phase functions. The model has also been evaluated against various
other models that participated in an international multi-model comparison for solar fluxes and
photolysis calculation (PhotoChem-2008 in Chipperfield et al., 2010) and against the
measurements from actinic flux spectroradiometers during the Atmospheric Tomography
(ATom) mission (Hair et al., 2018). In the aforementioned evaluations, the fast-JX model is
among the models with good performance. The model calculations provide three ratios: (i) CIdir,
the ratio of direct downward solar radiation at the surface (Rdir@srf) to the incoming total solar
radiation flux at the top of the atmosphere (Rtot@toa), (ii) CIdiff, the ratio of the downward
diffuse solar radiation flux (Rdiff@srf) to Rtot@toa, and (iii) CI, the ratio of total solar radiation
at the surface to Rtot@toa. Note that all Rs are for the 400-700 nm spectral band. Results for
different biomass burning AODs (including the clean air condition, where AOD = 0) for cloud-
free conditions are shown in Figure 9a.  When the sky is clear and clean (both cloud-free and
without aerosols), roughly 90% of the incoming solar radiation at the top of the atmosphere can
reach the plant canopy (i.e., CIdir + CIdiff ≈ 0.9 at BBAOD = 0). The direct solar flux decreases
rapidly as the atmosphere becomes polluted (i.e., as BBAOD increases), but for BBAOD levels
less than ~0.75, the diffuse solar flux increases. The two are equivalent at AOD ~ 0.5. This light
redistribution from direct to diffuse can significantly stimulate plant photosynthesis given that
plants use diffuse light more efficiently. Ecosystems could still respond positively to the increase
of BBAOD even if the incident diffuse radiation decreases below its peak value, though for some
value of BBAOD, the reduction in total radiation will be large enough to overwhelm the impact
of increased diffuse radiation, and plant photosynthesis will be lower than that for clean sky
conditions.

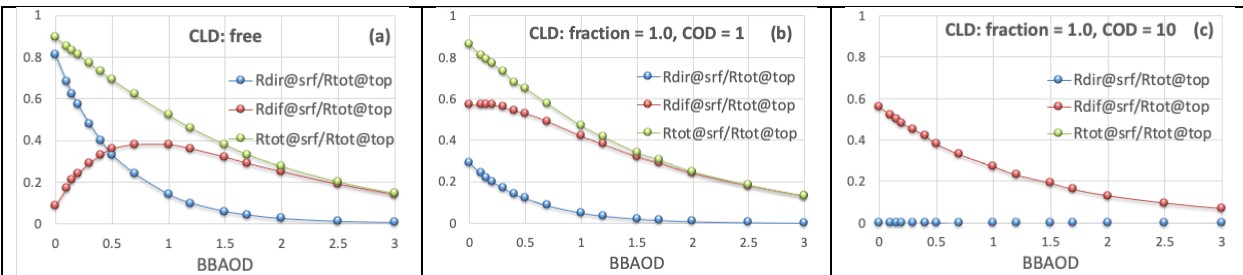

Figure 9. The ratio of Rdir@srf to Rtot@toa (blue), which presents the clearness index for the direct radiation portion (CIdir), the ratio of Rdiff@srf to Rtot@toa (red) for the diffuse radiation portion (CIdiff), and the ratio of Rtot@srf to Rtot@toa (green). Here, Rtot@toa is incoming total solar flux at the top of atmosphere (TOA), Rdir@srf is surface downward direct solar flux, Rdiff@srf is surface downward diffuse solar flux, and Rtot@srf is sum of Rdir@srf and Rdif@srf. All Rs are over 400-700 nm. 9a) the change of the radiative flux ratios in BBAOD = 0-3 under clear sky condition. 9b) same as left panel but under cloudy conditions (cloud fraction =1) with COD=1. 9c) same as middle panel but for COD=10. Calculations use fast-JX radiation model column version adopting a standard atmospheric condition of typical tropics at ozone column = 260 Dobson Units, SZA = 15°, and surface albedo = 0.1.

The Amazon dry season is characterized by high biomass burning aerosol loading combined with
low cloud cover, a good match to obtain more diffuse radiation without the loss of too much total
radiation. However, as we have pointed out, cloud impacts on radiation typically dominate those
of aerosols. To examine this, we repeated the radiation model calculations after adding, at the top
of the aerosol layer (~3.5km), a cloud layer with a cloud fraction of 1.0 and a cloud optical depth
(COD) of 1 (Figure 9b) and 10 (Figure 9c). The latter COD is close to the mean liquid cloud
COD over the Amazon dry season (Figure 3). The impact on Rdir@srf and Rdiff@srf is quite
large even with a very thin overhead cloud (Figure 9b). Without BBaer, the clouds already
produce abundant diffuse light that can reach the surface (i.e., CIdiff > 50%,  as seen in both
Figure 9b-c), while almost shutting down the direct light (i.e., CIdir < 1% in Figure 9c).
Accordingly, for full cloud coverage, a clean sky (i.e., no aerosols) would provide the best
conditions for plant growth. When fires start, the diffuse light declines rapidly, reducing the
potential for plant growth. At BBAOD ~ 3 the ratios among Figure 9a-c look similar, that is,
essentially very little radiation reaches the surface.
The simple examples in Figure 9 illustrate the complicated responses of direct and diffuse light
to the presence of aerosol and cloud. Measurements indicate that plant growth peaks for a
clearness index (CI, defined as CIdir+CIdiff) of about 0.4-0.7 for some forest ecosystems (Butt
et al., 2010, Letts and Lafleur, 2005). This CI range translates, based on Figure 9, to a BBAOD
range of about 0.3~1.5 in clear sky and 0~0.5 in cloudy-sky conditions.
**3.3 How the ecosystem responds to the BBaer diffuse radiation fertilization effect**

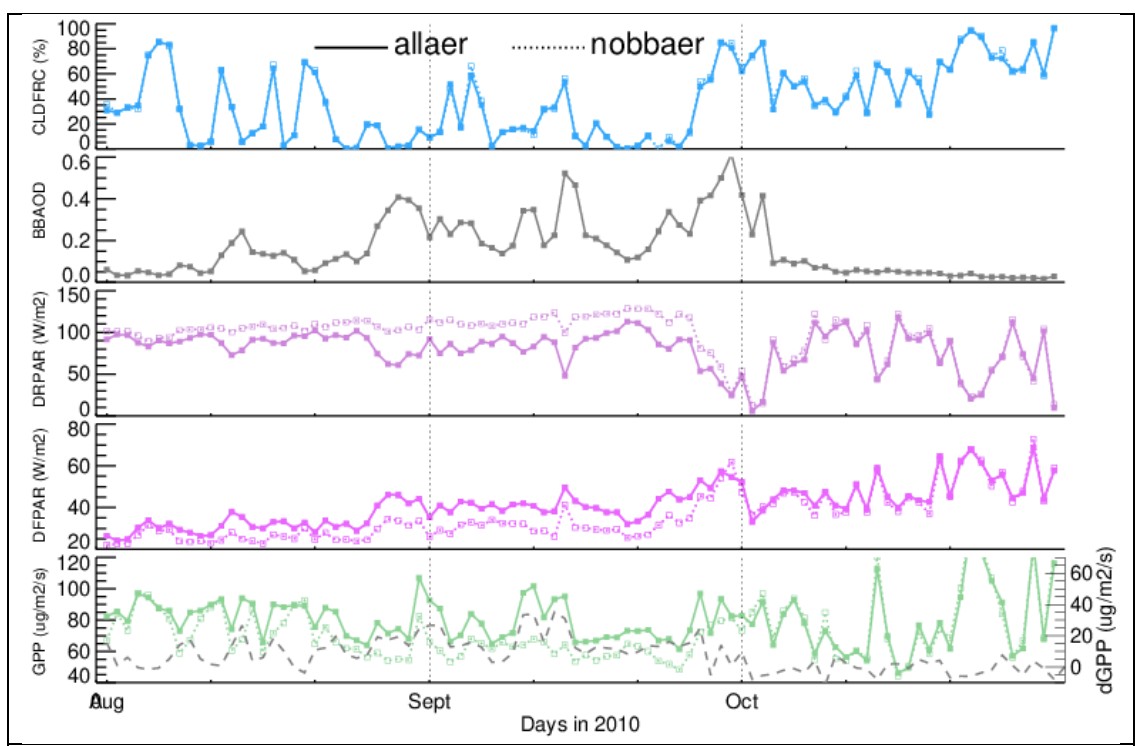

Figure 10. GEOS simulated daily values of total cloud fraction (CLDFRC, %), biomass burning AOD (BBAOD), direct PAR (DRPAR, Wm$^{-2}$), diffuse PAR (DFPAR, Wm$^{-2}$), and gross primary growth (GPP, μg/m$^2$/s) from the two experiments of pair1 at a selected site (54°W, 15°S; marked with a diamond in Figure 11) during Aug-Oct 2010. The grey dashed line in the bottom panel shows the absolute GPP difference (dGPP) between allaer and nobbaer.

We first examine the two experiments in pair1 by taking a close look at the time series of
aerosol, cloud, radiation, and ecosystem responses generated at a selected site (54°W, 15°S)
during Aug-Oct 2010 (Figure 10) (site location marked in Figure 11), with the aim of extending
the general understanding gained in section 3.2 to a real case study at a single site in the
Amazon. This is an interesting site and period, showing a large DFPAR change (Figure 11f) and
providing a wide variety of conditions for study – the sky alternates between clear and cloudy
conditions in August, is relatively clear in September but relatively cloudy in October, and the
biomass burning aerosols increase in August, peak in September, and diminish greatly in early
October (Figure 10). During August-September, when the atmosphere experiences biomass
burning pollution, the allaer (with BBAOD light fertilizer) and nobbaer (without BBAOD light
fertilizer) results differ significantly: DRPAR for allaer (solid line) lies below that for nobbaer
(dotted-line), while DFPAR and GPP for allaer are generally higher than those for nobbaer. In
October, the sky is almost clean (i.e., low BBaer), leading to very similar results for DRPAR,
DFPAR, and GPP between the two experiments. Looking closer, we see that the changes of
DRPAR, DFPAR, and GPP between allaer and nobbaer are more prominent when the
atmosphere has low cloudiness and high aerosol (e.g., at the end of August), confirming both that
BBaer does transform some of the direct light at the surface into diffuse light and that plants are
more efficient in their use of diffuse light. When both cloudiness and aerosols are high (e.g., at
the end of September), the influence of aerosols is overwhelmed by clouds, and the impact of the
aerosols on radiation and the ecosystem becomes secondary.
We now evaluate BB aerosol impacts on radiation and ecosystem fields over the Amazon during
August 2010, when the aerosol has its largest impact. Figure 11 shows the simulated Amazon
DRPAR, DFPAR, and GPP fields from the two experiments comprising pair1 (nobbaer and
allaer). The distribution of DRPAR shows a clear spatial gradient, with low values in the
northwest and high values in the southeast, and the spatial pattern of DFPAR shows the reverse
pattern. These features are primarily controlled by the cloud distribution (Figure 3). Comparing
the nobbaer and allaer results by calculating field relative change (i.e., (allaer-nobbaer)/allaer),
we find that BBaer decreases DRPAR by 16% and increases DFPAR by 10% over the Amazon
region, with maximum local changes of up to -50% for DRPAR and 25% for DFPAR.
Interestingly, these maxima are not co-located, though the spatial patterns of perturbations do
agree with each other. The mismatch in the locations of the maxima in the difference fields
implies a nonlinear response of direct and diffuse light to aerosol and cloud particles (see section
3.2). In response to the inclusion of BBaer, the Amazon GPP increases by 10%. That is, the
increase in GPP stemming from the increase in the diffuse light fraction overwhelms a potential
reduction in GPP from a reduction of total PAR.  When we consider all burning seasons over the
7-year studied period, the biomass burning aerosol increases DFPAR by 3.8% and decreases
DRPAR by 5.4%, allowing it to increase Amazon GPP by 2.6%. However, the 7-year averaged
GPP increases by 0.99% (Table 2), which is much less than the value during burning seasons.
We also examine the multi-year (2010-2016) BBaer impacts on net primary production (NPP),
that is, the rate at which carbon is accumulated (GPP) in excess of autotrophic respiration. In
essence, NPP can be considered a proxy for the net plant sink of atmospheric carbon. Figure 12
shows monthly and long-term averaged NPP over the Amazon Basin from the two experiments
comprising pair1. The monthly change of NPP (i.e., dNPP = NPP(allaer) – NPP(nobbaer)) is
shown in the figure as a green line. Each year, during the August-September period when BBaer
is high and cloudiness is low over the Amazon, BBaer is seen to enhance NPP. The percentage
difference of annually-averaged NPP (dNPP/NPP(nobbaer)*100) in % is 4.2, 0.06, 1.9, 0.5, 1.3,
1.9, and 1.0 for the seven studied years.  That means the BBaer-induced NPP increases range
from 5 TgC yr$^{-1}$ or 0.06% (2011) to 278 TgC yr$^{-1}$ or 4.2% (2010), with a seven-year average of
92 TgC or 1.5%. This is equivalent to storing 92TgC annually within the Amazon ecosystem
during the studied period. The $CO_2$ fire emission data from the GFED4.1s emission inventory
indicate that over this area and time period, fires emit ~250TgCyr$^{-1}$. The NPP enhancement due
to the BBaer-induced diffuse sunlight fertilization thus compensates for about 37% of carbon
loss by fires.

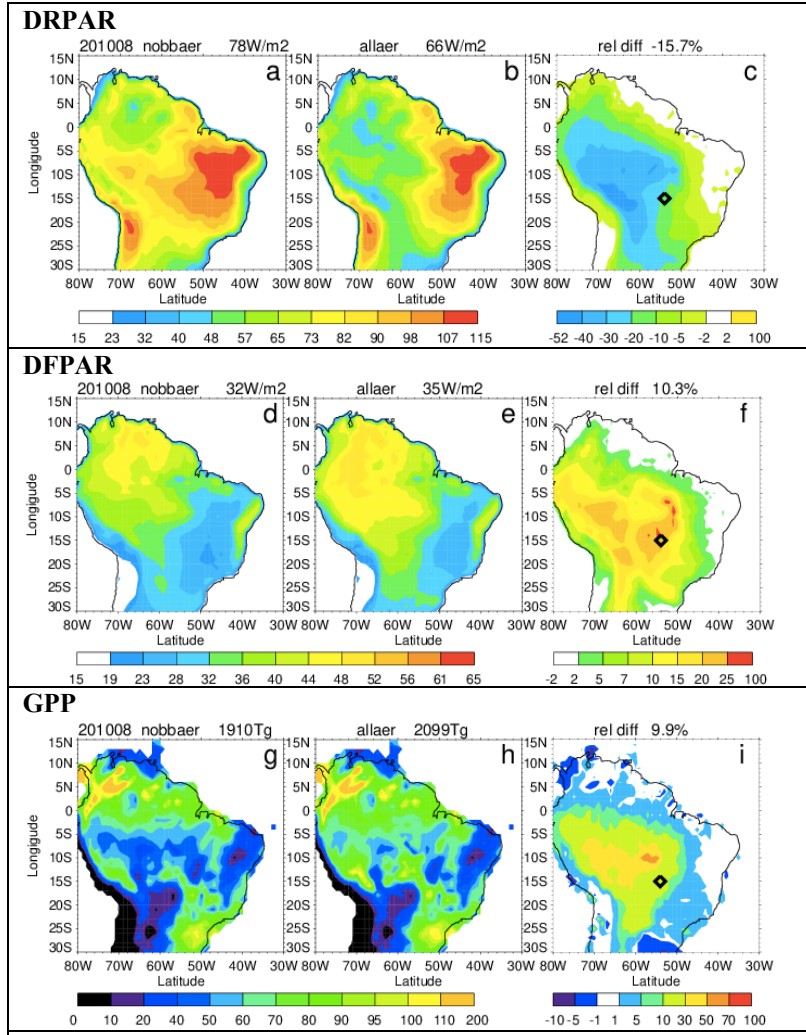

Figure 11. August 2010 Amazon DRPAR (W m$^{-2}$) (a, b, c), DFPAR (W m$^{-2}$) (d, e, f), and GPP (kg m$^{-2}$ s$^{-1}$) (g, h, i) from the nobbaer (a, d, g) and allaer (b, e, h) GEOS experiments. The (c, f, i) shows the relative change between allaer and nobbaer. All values are the Amazon regional average except the GPP values of (g, h) are regional total. Further analyses on the (c, f, i) diamond locations are given in Figure 10.

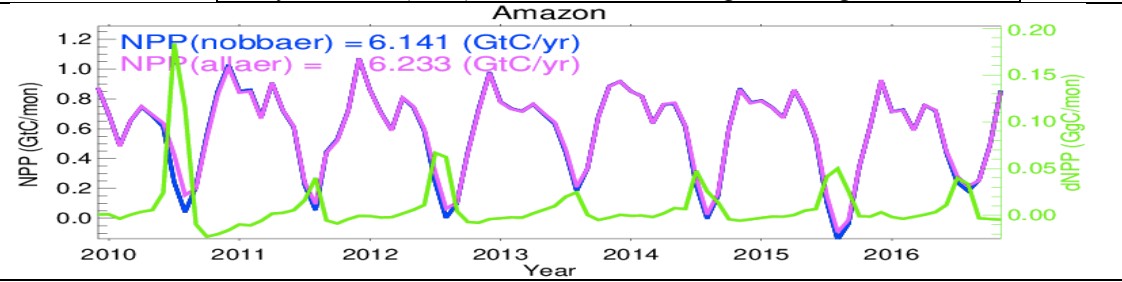

Figure 12. Monthly net primary product (NPP) over the Amazon basin (i.e. the land area of 80W-30W, 25S-5N) for the allaer (magenta-solid) and nobbaer (blue-solid) GEOS simulations with the multiyear mean values indicated in the legend. The monthly difference of NPP (dNPP = NPP(allaer) – NPP(nobbaer)) is shown by the green line and the right y-axis.

To assess how our simulated GPP/NPP response compares with other existing model estimates,
we summarize all relevant studies in Table 2. In addition to differences in model formulations of
fundamental physical mechanisms, these studies also differ in model simulation configuration
(e.g., online vs offline, freeGCM vs Replay), BB emission inventory, and study period. Although
our estimates of the increases in NPP across the Amazon region have a wide interannual
variation (ranging from 0.5 to 4.2%), our 7-year averaged NPP increase (1.5%) is close to the
value (1.4%) reported by Rap et al. (2015). Both studies considered only aerosol DRFE with
cloud presence. The NPP can be increased up to 52% in the burning season under clear-sky
conditions (Moreira et al., 2017). By accounting for the feedback from aerosol-climate
adjustments, the influence of aerosol on GPP/NPP is further increased (Malavelle et al., 2019;
Strada et al., 2016).
Table 2: Summary of model estimation of GPP increase in response to biomass burning aerosol
over Amazon Basin

| Study | This work | Malavelle2019 | Moreira2017 | Rap2015 | Strada2016 |
|---|---|---|---|---|---|
| GPP | 1.0% (dir+dif) | | 27% (dir+dif) | 0.7% (dir+dif) | 3.4% (dir+dif+clm)) |
| NPP | 1.5% (dir+dif) | 1.9 to 2.7% (dif+dir+clm) 1.5 to 2.6% (dif) -1.2 to -2.5% (dir) 1.6 to 2.4% (clm) | 52% (dir+dif) | 1.4% (dir+dif) | |
| Period | Annual average over 2010-2016 | Annual average over 30 model years, 2000 climate, | Sept., 2010 under cloud-free condition | Annual average over 1998-2007 | Annual average over 30 model years, 2000 climate |
| Atmospheric Model | GEOS ESM | HadGEM2-ES | BRAMS | | ModelE2 ESM |
| Running mode | replay | freeGCM | Regional model with ICBC from NCEP | offline | freeGCM |
| Vegetation model | Catchment-CN (using LSM4 for photosynthesis) | JULES | JULES | JULES | YiBs |
| Radiation model | RRTMG_SW | SOCRATES | CARMA | A two-stream radiative transfer model (Edwards and Slingo, 1996) | k-distribution approach with various updates (Schmidt et al., 2014) |
| Cloud model | Cloud microphysics model (Barahona et al., 2014) | | | Monthly mean clouds from ISCCP-D2 | a mass flux cumulus parameterization (Del Genio and Yao, 1993) |
| Aerosol model | GOCART | CLASSIC | CCATT | GLOMAP | OMA |
| BB emission | GFED4s | GFEDv2 1997-2006 average | 3BEM | GFED3 | IPCC AR5 |

dir, dif, and clm represent for direct radiation, diffuse radiation, and climate adjustment, respectively
3BEM: the Brazilian Biomass Burning Emission
BRAMS: Brazilian developments on the Regional Atmospheric Modeling System
CARMA: the Com-munity Aerosol and Radiation Model for Atmospheres
CCATT: a Eulerian transport model suitable to simulate trace gases and aerosols
CLASSIC: the Coupled Large-scale Aerosol Simulator for Studies In Climate
GLOMAP: The 3-D GLObal Model of Aerosol Processes Model
HadGEM2-ES:  The Hadley Centre Global Environment *Model*, version 2-Earth System
IPCC AR5: The Intergovernmental Panel on Climate Change Fifth Assessment Report
ISCCP-D2: the International Satellite Cloud Climatology Project
JULES: the Joint UK Land Environment Sim-ulator v3.0
OMA: One-Moment Aerosol,
SOCRATES: Suite Of Community RAdiative Transfer codes based on Edwards and Slingo
YIBs: The Yale Interactive Terrestrial Biosphere model
**3.4 How clouds adjust the BBaer diffuse radiation fertilization effect**
Our second objective in this study is to investigate how the presence of clouds modulates the
ability of BBaer to affect GPP. We highlight the cloud impact because even at the same biomass
burning aerosol optical depth (BBAOD), the surface downward DRPAR and DFPAR can be
very different between cloudy and cloud-free conditions (see section 3.2). As mentioned above,
the Amazon's so-called "dry season" still features a considerable amount of cloud, and the
cloudiness levels vary significantly from year to year. This raises some questions: How do
clouds affect the aerosol impact on radiation fields during the Amazon biomass burning season?
Could different levels of background clouds have different impacts on the efficacy of the BBaer
DRFE? There are two distinctive features in clouds and aerosols that require us to treat them
differently in their impact on the radiation flux to the ecosystem. First, like our distinction of
natural and anthropogenic aerosols in their impact on air quality and climate, the cloud is a more
natural phenomenon, while biomass burning aerosols (BBaer) can be, at least partially,
controlled by humans. Second, clouds are much more efficient in controlling both direct and
diffuse radiation fields than aerosol (Figure 6). What is the potential range of the variation of
Amazon clouds in burning seasons when the Amazon experiences environments of La Niña,
normal years, and El Niño? To what extent does this range of cloud variation adjust the
efficiency of "diffuse radiation fertilization effect" under the same emission strategy? These
questions were not addressed clearly in previous studies, and we have tried to answer these
questions in this study. Here, to quantify the cloud influence, we examine BBaer impacts during
clear-sky (cloud cover < 0.1), cloudy-sky (cloud cover 0.1-0.3, 0.3-0.6 and >0.6), and all-sky
conditions based on GEOS gridded daily cloud cover over the Amazon region as shown in
Figure 13.
Generally, the curves for BBAOD (solid black line) and dGPP (dashed light-blue line) are
strongly and positively correlated, from R = 77.4% for cloud cover > 0.6 (Figure 13d) to R >
94.5% for the four other cloudiness conditions (Figure 13a-c, e). This indicates that interannual
changes in dGPP are primarily controlled by interannual fluctuations of biomass burning
aerosols. The correlation presumably stems from the fact that biomass burning aerosols increase
the diffuse PAR reaching the canopy (dashed pink line) although they decrease the total PAR
(dotted purple line) via decreasing direct PAR (Table 3 and Table S1a). This aerosol-radiation-
GPP relationship is seen to vary with cloud amount with clouds acting to reduce the aerosol
impact; both the diffuse radiation and the GPP show larger changes with BBAOD under clear
sky conditions. The overall (i.e., all-sky) aerosol impact on dGPP is similar to that for a cloud
coverage of 0.3-0.6, simply because the averaged cloud coverage over the Amazon during the
studied period is roughly in that range.

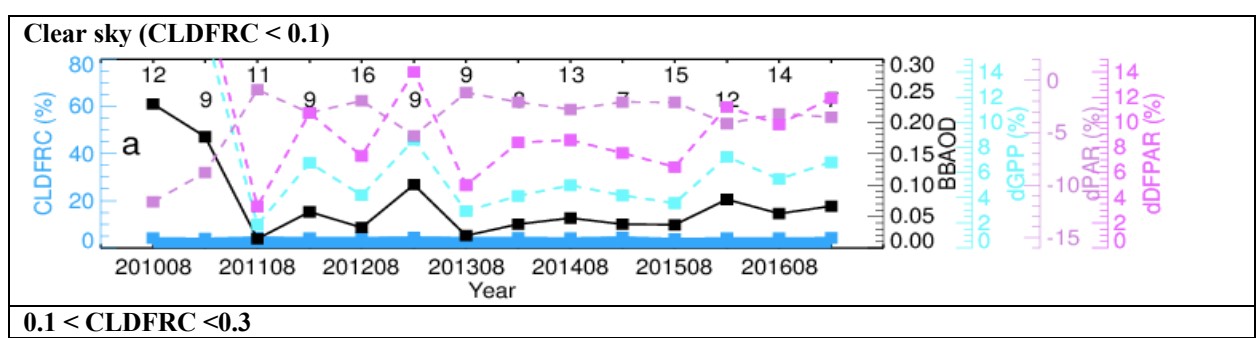

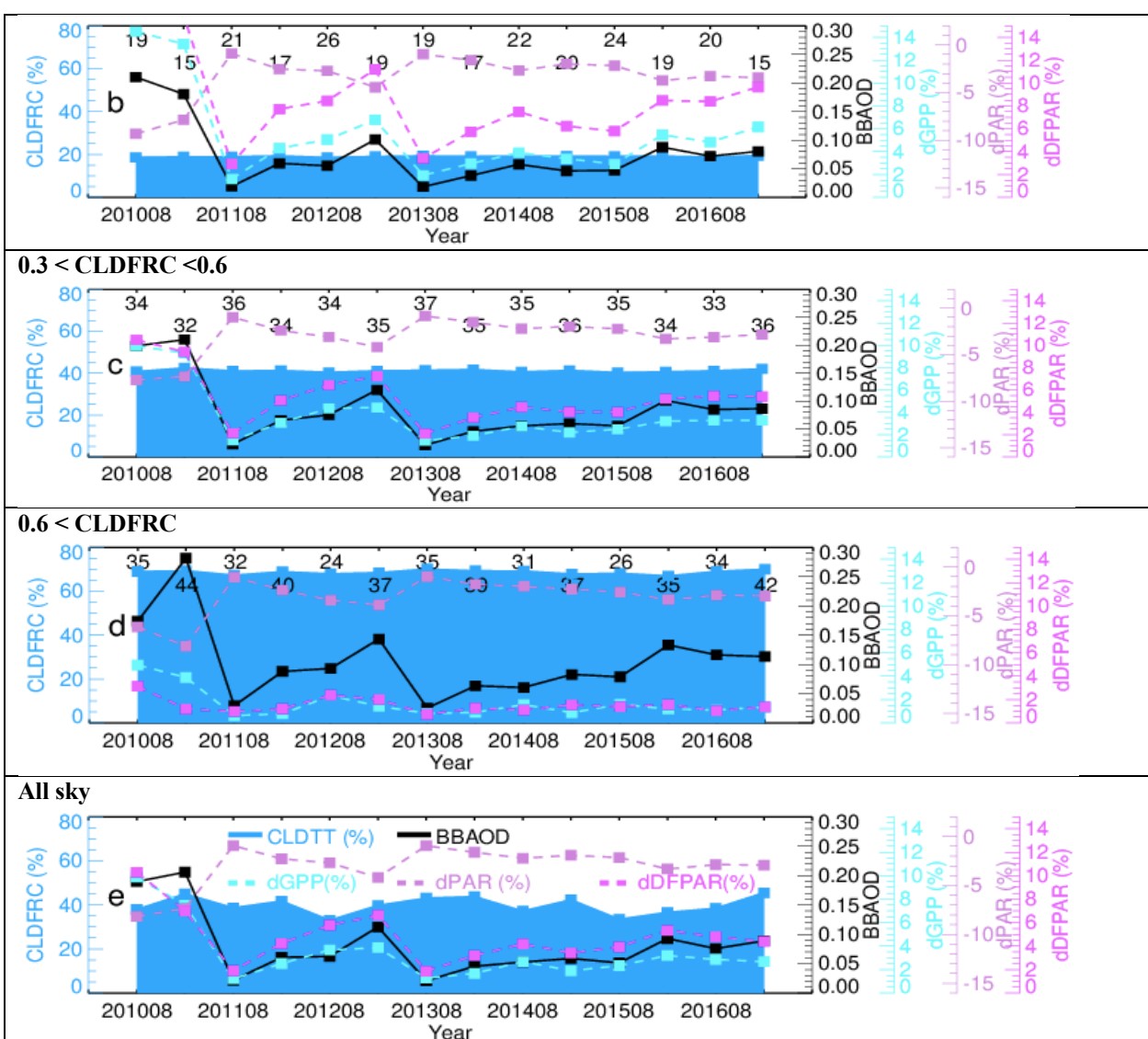

Figure 13. Monthly (August and September) averaged fields during 2010-2016 over Amazon range (80W-30W, 25S-5N) for different cloudy conditions. The fields shown here are CLDFRC (shaded area), biomass burning aerosol optical depth (BBAOD, black solid line), and the changes of GPP (dGPP), direct (dDRPAR) and diffuse (dDFPAR) fields due to biomass burning aerosol impact on radiative fields (dashed lines) estimated by the two pair1 experiments. Note all the changed fields are calculated as dX (%) = (X(aller)-X(nobbaer))/X(nobbaer)*100.0, here X = GPP, DRPAR, or DFPAR. The numbers marked in (a)-(d) are the frequency of occurrence in % of the corresponding cloud cover over the Amazon basin in each month. Note the dGPP is 119.5% (201008) and 92.6% (201009) in (a). The dDFPAR is 111.1% (201008) and 105.5% (201009) in (a) and 97.1% (201008) in (b).


Figure 13 and Table S1e show that on an interannual (dry season) basis, the aerosol DRFE
differed the most between 2010 and 2011 (i.e., the dGPP was 8.7% in 2010 and 1.8% in 2011).
During these two years, the average cloud fractions (CLDFRC) are similar, 42% in 2010 and
41% in 2011, but BBAOD decreased significantly, by about 80% from 0.198 in 2010 to 0.042 in
2011. Thus, although cloudiness does temper the impact of aerosols on radiation and the
ecosystem, the interannual variation of the aerosol DRFE is primarily controlled by variations in
biomass burning aerosols (e.g., > 6 times variation of biomass burning emissions and BBAOD,
table S1e).  In addition to the detailed information given in Tables S1a-e and S2a-e, we
summarize in Table 3 the averaged GPP, DFPAR, DRPAR, CLDFRC, and BBAOD during Aug-
Sept, 2011-2016 over the Amazon region in all-sky conditions. Also given in Table 3 is the
multi-year (2011-2016) averaged GPP over the Amazon region from all four simulations.
Table 3. Summary of mean GPP, DRPAR, DFPAR, CLDFRC and BBAOD over Aug-Sept of
2011-2016, as well as the relative changes of GPP, DRPAR, DFPAR and CLDFRC within a pair
of simulations.

| pair | experiment | GPP | DRPAR | DFPAR | CLDFRC | BBAOD |
|---|---|---|---|---|---|---|
| | | GtC/Amazon | $Wm^{-2}$ | $Wm^{-2}$ | | |
| Pair1 | allaer | 1.88 | 72.5 | 36.8 | 0.395 | 0.062 |
| | nobbaer | 1.84 | 76.5 | 35.3 | 0.395 | |
| | Diff (%) | 2.5 | -5.3 | 4.1 | 0 | |
| Pair2 | callaer | 1.96 | 64.5 | 38.0 | 0.396 | 0.212 |
| | cnobbaer | 1.83 | 75.4 | 35.1 | 0.395 | |
| | Diff (%) | 6.9 | -14.4 | 8.2 | 0 | |

Recall, the pair2 experiments are equivalent to the pair1 experiments except for using the 2010
BB emissions for every year during 2011-2016.  By jointly analyzing pair1 and pair 2, we can
quantify the impacts of two different sets of BB emissions during the study period. This is, in
principle, similar to the method of aerosol radiative forcing (RF) estimation (i.e., estimating
aerosol radiative effect (RE) with and without aerosol for present-day (pair1) and pre-industrial
(pair2)  conditions and then deriving RF as a difference between the two pair REs). Here we
study the sensitivity of the aerosol DRFE to a unit change of AOD. We call it susceptibility of
the DRFE to BB aerosols. That is, on a daily basis, the sensitivity of a variable X to a change in
the biomass burning AOD is calculated as: $ddX/dAOD = ((dX)_1-(dX)_2)/(AOD_1-AOD_2)$. Here, the
X represents GPP, DRPAR, and DFPAR, and the subscripts 1 and 2 represent the pair1 or pair2
experiment, respectively.
ddX/dAOD is computed on a gridded daily basis over August-September of 2011-2016. The
calculations are then catalogued according to daily cloud cover fraction – we combine the results
within each of 10 cloud fraction bins (0-0.1, 0.1-0.2, …, 0.9-1.0). To examine the maximum
impact of interannual cloud change during our study period, the binned ddX/dAOD vs. CLDFRC
relationship is also computed separately from daily (August-September) values in 2013 and from
corresponding daily values in 2015, as these are the years for which monthly cloud cover is
around the maximum (0.44) and minimum (0.35), respectively (Figure 13 and table S1e).
Figure 14 shows the results. An almost linear relationship is seen between the ddX/dAOD values
and cloud cover fraction. BB aerosols increase GPP in clear sky conditions (e.g., 29.6 $kgm^{-2}s^{-1}$)
but decrease it under full cloudiness conditions (e.g., -5.8 $kgm^{-2}s^{-1}$). The cloud fraction at which
BB aerosol switches from stimulating to inhibiting plant growth occurs at ~0.8. Cloud conditions
thus not only affect strongly the strength of the aerosol DRFE but can also change the
fundamental direction of the effect. The lines produced for the three different study periods are
fairly similar, indicating that the relationship of ddX/dAOD to CLDFRC is fairly stable within
the range of cloud cover seen over the Amazon during the period of interest. Figure 14 also
indicates that the dGPP can change from 18.5 to 15.5 ($kgm^{-2}s^{-1}$) with a unit AOD of burning
particles released to the atmosphere under the range of Amazon interannual cloud variation in
dry season, which is 0.35 to 0.44 in our study period. In other words, there is ~20% dGPP
uncertainty adjusted by background Amazon cloud. Our work demonstrates quantitively the role
of clouds in tempering the aerosol diffuse radiation fertilization effect.

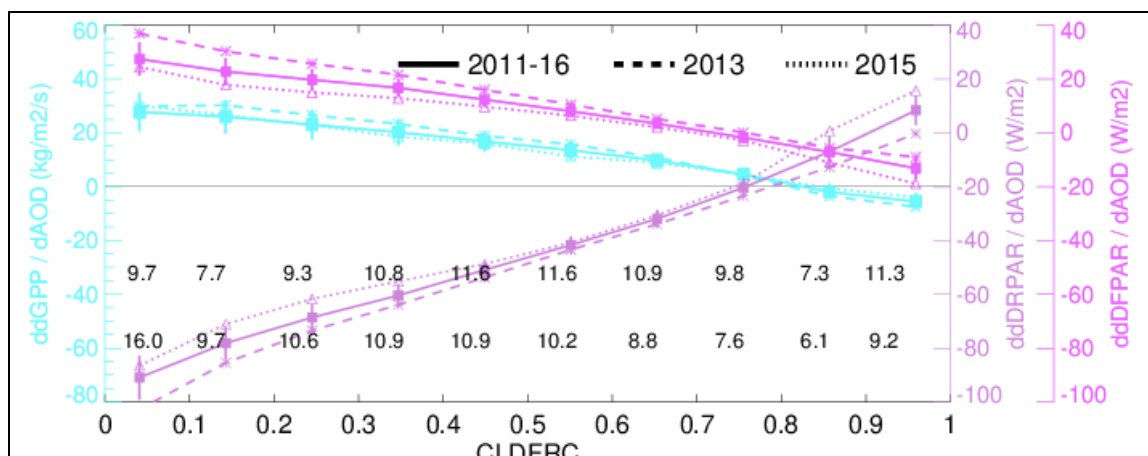

Figure 14. Radiation (DRPAR and DFPAR) and ecosystem (GPP) perturbation on every unit AOD change calculated combining the two pairs of experiments, i.e. $(dGPP_1-dGPP_2)/(AOD_1-AOD_2)$, $(dDRPAR_1-dDRPAR_2)/(AOD_1-AOD_2)$, and $(dDFPAR_1-dDFPAR_2)/(AOD_1-AOD_2)$, here subscripts referring to the experiments of pair1 and pari2. These changes are sorted out based on the values of grid box cloud fraction on a daily basis during the reported timeframe (e.g., solid-line for Aug-Sept, 2011-2016, dash-line for Aug-Sept 2013, and dot-line for Aug-Sept 2015). Also shown are the vertical bars for one standard deviation and the number of the occurrence frequency in % of each cloud fraction bin (0.1 increment) over the Amazon region for 2013 (first row) and 2015 (second row).

**4. Conclusions**
We use the NASA GEOS ESM system with coupled aerosol, cloud, radiation, and ecosystem
modules to investigate the impact of biomass burning aerosols on plant productivity across the
Amazon Basin under the natural background cloud fields experienced during 2010-2016 – a
period containing a broad range of cloudiness conditions. We find that the biomass burning
aerosol DRFE does stimulate plant growth and has a notable impact on Amazon ecosystem
productivity during the biomass burning season (August-September). In the long-term mean, the
aerosol light fertilizer increases DFPAR by 3.8% and decreases DRPAR by 5.4%, allowing it to
increase Amazon GPP by 2.6%. On a monthly basis, the DRFE can increase GPP by up to 9.9%.
Consequently, biomass burning aerosols increase Amazonia yearly NPP by 1.5% on average,
with yearly increases ranging from 0.06% to 4.2% over the seven years studied. This 1.5% NPP
enhancement (or ~92TgC yr[-1]) is equivalent to ~37% of the carbon loss due to Amazon fires.
The aerosol DRFE is strongly dependent on the presence of clouds, much stronger in clear sky
conditions and decreases with the increase of cloudiness. A fairly robust linear relationship is
found between cloud cover fraction and the sensitivity of radiation and GPP change to a change
in biomass burning AOD. BB aerosols stimulate plant growth under clear-sky conditions but
suppress it under full cloudiness conditions. Over the Amazon region within our study period,
the cloud fraction at which a unit AOD switches from stimulating to inhibiting plant growth
occurs at ~0.8. Note, however, that while our results show a clear sensitivity of the aerosol
DRFE to cloudiness, interannual variations in the aerosol light fertilizer's overall effectiveness
are controlled primarily by interannual variations in biomass burning aerosols during our studied
period because biomass burning AOD can vary by a factor of 6 from year to year. The associated
large variations in BBAOD are inevitably propagated to the radiation and ecosystem fields.
Overall, our work indicates that feedbacks between aerosols, radiation, and the ecosystem need
to be performed in the context of an atmospheric environment with a cloud presence.
This study examines the potential for the biomass burning aerosol DRFE to stimulate growth in
unburned forest over the Amazon basin. The net feedback of Amazon fires on the Amazon
biome is still an open question. Some changes, such as increasing atmospheric $CO_2$ and aerosols,
serve as forest fertilizers, whereas others, such as increasing $O_3$ pollution levels and the
deposition of smoke particles on plant leaves, reduce plant photosynthesis. On top of this, fires
also induce changes in meteorological fields (e.g., temperature, precipitation, clouds) that can
affect plant growth (Malavelle et al., 2019; Strada and Unger, 2016; Unger et al., 2017). More
efforts are needed to investigate the ecosystem effect of Amazon fires by integrating all these
potential factors.

**Acknowledgements:**
The authors thank the various observational groups (i.e., AERONET, CERES-EBAF, FluxCom,
FluxSat, and GoAmazon). HB and MC was supported by the NASA ACMAP funding (no.
NNX17AG31G). PRC was supported by the Chemistry-Climate Modeling workpackage funded
by the NASA Modeling, Analysis, and Prediction program (David Considine, program
manager). JES was supported by the by the U.S. Department of Energy's Atmospheric System
Research, an Office of Science Biological and Environmental Research program; PNNL is
operated for the DOE by Battelle Memorial Institute under contract DE-AC05-76RL01830.
Resources supporting this work were provided by the NASA GMAO SI-Team and the High-End
Computing (HEC) Program through the NASA Center for Climate Simulation (NCCS) at
Goddard Space Flight Center (GSFC). FluxSat data were provided by Joanna Joiner group in
GSFC. GoAmazon data were obtained from the Atmospheric Radiation Measurement (ARM)
user facility, a US Department of Energy (DOE) Office of Science User facility managed by the
Biological and Environmental Research program.

**Data Availability:**
All of the observational data used in this study are publicly accessible, e.g., AERONET
(https://aeronet.gsfc.nasa.gov), CERES-EBAF (https://ceres.larc.nasa.gov/data/), FluxCom
(http://www.fluxcom.org), FluxSat (https://avdc.gsfc.nasa.gov), and GoAmazon
(https://www.arm.gov/research/campaigns/amf2014goamazon). The GEOS model results can be
provided by contacting with the corresponding author.

**Author contributions:**
H.B. took an overall responsible for the experiment design, model simulation, and data analysis.
E.L., R. D. K., S. P. M., and F. Z. contributed to the ecosystem study, D. O. B. contributed to the
cloud study, M. C., P. R. C., A. S. D, M. E. M., and H. Y. contributed to the aerosol study and
the model-observation comparison, P. N. contribute to the radiation study, and J. S. provided the
GoAmazon results. All authors contributed to the paper writing.

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
