# Peer review of "The Response of the Amazon Ecosystem to the Photosynthetically Active"

_Atmospheric Chemistry and Physics, 2021_

## Author Comment (AC1)

**Review of "The Response of the Amazon Ecosystem to the Photosynthetically Active Radiation Fields: Integrating Impacts of Biomass Burning Aerosol and Clouds in the NASA GEOS ESM"**

**by Bian et al.**

**Summary:**

There has been a growing number of studies looking at the potential impact on land carbon uptake from increased availability of diffused radiation associated with aerosol particles. Although conceptually simple, this effect is hard to quantify accurately as complex couplings between different components of the Earth system are at play. An Earth System Modelling (ESM) approach appears as a natural framework for this kind of problem, yet only a handful of studies using ESMs has been published so far. This submission from Bian et al. is therefore timely as they used the results from simulations performed with the NASA GEOS-ESM to analyze the impact of biomass burning aerosols on the Amazon rainforest gross primary productivity. The diffuse light fertilization effect from aerosols is not only uncertain, but it is also buffered by clouds as those compete with aerosols for radiation. This is in a way similar to pre-industrial natural aerosols controlling the amplitude of the anthropogenic aerosol radiative forcing. The role of clouds on the aerosol diffuse light fertilization effect has not been properly explored before and Bian et al. provide a novel quantification for this modulating effect.

It's an interesting paper, easy to read, the structure is sound, and the supporting material is adequate. The work perfectly fits within the topics covered by ACP so I strongly support its publication after addressing those minor few points.

We thank the reviewer for his/her good summary and comments. Please see our point-to-point reply below in blue.

**General Comments:**

1. How was the 7 years period selected? Surely the observational datasets used for the evaluation cover a longer range so it could have been an opportunity to extend the statistics.

   We intended to select years that were continuous and covered the periods of La Niña, normal years, and El Niño. This 7-year period (2010-2016) fit our

selection standard according to the NOAA's Ocean Niño Index (ONI) anomaly (see Figure S1 and text lines 317-320).

[Figure]

Figure S1. Ocean Niño Index (ONI) anomaly, Jan 2004-July 2019. Source: https://origin.cpc.ncep.noaa.gov/. El Nino or La Nina is identified when the ONI anomaly exceeds ±0.5°C for at least five consecutive months. The study period (2010-2016) covers La Niña (2010-2011), normal (2012-2014), and El Niño years (2015-2016).

We also added the information of long-term BB OA emissions (i.e., 1997-2016) and long-term MERRA2 cloud fraction anomalies (i.e., 1995-2018) in Figure S2. The selected 7-year period represents well the long-term period in terms of the variation of BB emissions and cloud coverage. See lines 320-323.

[Figure]

Figure S2. (a) Monthly GFED4.1s biomass burning OC emission during 1997-2016 over the Amazon region. The year 2010 has a second largest emission and 2013 has a second lowest emission during the years when GFED4.1s has emission record. (b) MERRA2 monthly cloud fraction anomalies over the Amazon region during 1995-2018. The cloud fraction was highest in December 2010 and lowest in July 2012.

To examine whether the 7-year analyses are valid over a longer period, we also added a comparison of mean GPPs and their difference during ASO over two

multi-year periods (2010-2015 and 2003-2015) using the FluxCom and FluxSat observational data (Figure R1). The longer period covers all the years when observational data are available. Compared to the perturbed GPP due to aerosol diffuse radiation fertilization effect (dGPP(aer)) shown in Figure 11, the large dGPP shown here (Figure R1) appears in the areas outside the effective regions of dGPP(aer) (Figure 11).

[Figure]

Figure R1. Mean GPPs and their difference during ASO over two multi-year periods (2003-2015 and 2010-2015) using the FluxCom (a,b,c) and SatCom (d,e,f) observational data.

2. I would be tempted to rename the "fertilizer effect" as the "fertilizing effect" as fertilizer and effect are both nouns and it sounds odd to me. I'm not a native English speaker thought.

Based on the comments of both reviewers, we now use "diffuse radiation fertilization effect" throughout the manuscript.

3. How is the seasonality (not just August – September) of clouds in your simulations - e.g. ICTZ temporal/spatial position which would affect moisture availability - when compared against observations? Same question for the seasonality of GPP/NPP. Have you evaluated the simulated seasonal cycle against the two products mentioned in the manuscript (i.e. FluxCOM and FluxSat)? Those could be potentially be added to the supplementary material.

Based on the reviewer' suggestion, we have added Figures S5a-c and S7. Figure S5a-c, combining with Figure 3, shows seasonality of clouds from MODIS and GEOS during 2010-2016. The model has better performance during the period of Aug-Oct, which is the focus period of this study since Amazon fires occur periodically every year in this season. We added this discussion in lines 469-472. Figure S7 shows seasonality of GPP from FluxCOM, FluxSat and GEOS during 2010-2015. During all four seasons, regional FluxCom GPP is the lowest and FluxSat GPP is the highest. All observational and model GPPs show higher values during Nov-Apr than those during May-Oct. See lines 501-509.

4. The description of the representation of light interception by vegetated canopies (Line 235-237) could benefit from being described in a bit more details as this part of the model is central to the present study. Is it the same parameterization as the one in CLM4?

   Yes. The light interception by vegetation in this study adopts the same parameterization as the one in CLM4. We modified the relevant description in lines 235-236 by explicitly indicating this point: "The light interception by vegetations in the GEOS Catchment-CN adopts the same parameterization as the one in CLM4. The photosynthesis and transpiration depend non-linearly on solar radiation. The canopy is assumed to consist of sunlit leaves and shaded leaves, and the DRPAR and DFPAR absorbed by the vegetation is apportioned to the sunlit and shaded leaves as described by Thornton and Zimmermann (2007). The prognostic carbon storages underlying the phenological variables are computed as a matter of course along with values of canopy conductance that reflect an explicit treatment of photosynthesis physics."

5. A bit off topic but I am curious as you mentioned photolysis and brown carbon developments in section 2. Are these two coupled? If so, is there any detectable impact on photolysis rates and consequently on ozone formation?

   No. The newly developed brown carbon absorption has not been implemented in the GEOS online photolysis simulation that may impact ozone chemistry. The new stronger brown carbon absorption in UV spectrum has only been accounted for in the GEOS online simulation of radiation. The photolysis calculation that demos the concept of how aerosol and cloud impact on surface downward radiation (Figure 9) was conducted with a stand-alone FastJX model.

6. If I understand the experimental design correctly, the radiation driving the atmospheric model in callaer and cnobbaer for pair 2 (respectively allaer and nobbaer for pair 1) is the same, namely R1. Does that imply that there is

potentially an implicit accounting of BB aerosol semi-direct effects in cnobbaer (respectively nobbaer), meaning that any potential change in cloudiness when doing a Diff between callaer and cnobbaer won't be captured? Given that 2010 was a drought year with high BB emissions (see figure 10), could this lack of change in CF be meaningful and affect the slope of ddGPP/dAOD?

Excellent point. Cloudiness (CF) and other relevant important meteorological fields have certain differences between the two pair simulations due to R1 associated with different aerosols or even between two simulations within a pair due to land ecosystem feedback. To examine whether these differences result in a meaningful impact on our conclusion, we compared CF, Skin temperature, and soil moisture simulated by all simulations (allaer and nobbaer in pair 1 and callaer and cnobbaer in pair2) over our study period, see Figures S3-4. The very small changes of these fields among the four simulations shown in the figures support our approach. The similar atmospheric fields stem from the model feature we adopted in this study. In order to focus strictly on the ecosystem response to the into-ecosystem light perturbed by biomass burning aerosols, we use a Replay mode configuration in simulations. In Replay mode, every six hours, the model atmospheric dynamic state (winds, pressure, temperature, and humidity) is set to the balanced state provided by the Modern-Era Retrospective analysis for Research and Applications, Version 2 (MERRA2) and then a six-hour forecast is performed until the next analysis is available. MERRA2 is the assimilation system that enable assimilation of modern hyperspectral radiance and microwave observations, ozone profiles observations, along with GPS-Radio Occultation datasets. Simulation with Replay is important in this study with two advantages: 1. Nudges GEOS dynamic fields to MERRA2 reanalysis ensuring atmospheric conditions of the four simulations are close to each other, therefore, resulting in more focus on the study of into-ecosystem radiative impact, 2. Ensures the observational-constrained meteorological fields used in our study. Please see lines 309-317.

7. Another aspect of the experiment design that is not clear to me is whether the inputs (not just radiation) from the atmospheric model to Catchement-CN in nobbaer and cnobbaer are from the atmosphere that has experienced R1 or from the atmosphere that has experienced R2. If the former, as I understand it, it would mean that the vegetation only 'feels' the aerosols via the change in dir/diff radiation but not via other changes in the energy balance that aerosols also introduce. Can you clarify?

The reviewer is right that all atmospheric calculations use R1, see Table 1. The experiments in this study were intentionally designed in this way so that the vegetation only 'feels' the aerosols via the change in dir/diff radiation but not via

other changes in the energy balance that aerosols also introduce. Future study may extend to include these other changes. We added this clarification in lines 366-370.

**Specific Comments:**

1. Line 29, change "the impact" to "this impact".

   Done.

2. Line 32, change "call" to "called". I would however argue that the light fertilization effect is only one of the impacts (plural, as in the manuscripts) resulting from aerosol radiative effects (e.g. less surface radiation affects the energy balance hence has an impact on vegetation productivity too). Maybe this sentence could be slightly reworded.

   The sentence has been changed to "The direct radiative impact of biomass burning aerosols on ecosystem productivity—called here the aerosol diffuse radiation fertilization effect—is found to increase Amazonian Gross Primary Production (GPP) by 2.6% via a 3.8% increase in diffuse PAR (DFPAR) despite a 5.4% decrease in direct PAR (DRPAR) on multiyear average during burning seasons."

3. Line 38, Replace "lost" by something like "average loss" otherwise it becomes slightly ambiguous (i.e. lost would correspond to the total loss over 7 years, so ~7 times 250-300 TgC/yr).

   Changed "equivalent to ~37% of the carbon lost" to "equivalent to ~37% of the average carbon lost".

4. Line 39, replace "is highest for" by either "is higher in" or "is at the highest in"

   Changed to "is the highest in".

5. Line 50, remove "dioxide", carbon has been sequestrated in different molecular forms by the vegetation.

   Done.

6. Line 59, "It is in the dry season, when light becomes a key-controlling". Could be reworded. Light is a key controlling factor outside of that period as well, but it is probably less of a bottleneck.

   Changed the sentence to "It is in the dry season, when more light reaches the canopy level, that the Amazon forest thrives."

7. Line 100, technically they accounted for the variability in cloudiness as those were free running simulations. However, the authors have not explicitly quantified the impact of this variably on the aerosol light fertilisation effect.

   Changed the sentence to "However, the authors have not explicitly quantified the impact of Amazon background clouds and their interannual changes in tempering the aerosol diffuse radiation fertilization effect (DRFE)."

8. Line 108, The definition of CI is not clear. Does it correspond to the ratio of total (i.e. dif + dir) light at surface over the total light at ToA ? Please clarify.

   The definition has been changed to "the ratio of total (i.e., direct plus diffuse) light at surface to the total incoming light at top of atmosphere".

9. Line 129, "sunlight … drenches the trees due to reduced rain". If by "drenches" you mean "floods the canopy with light", I would use a different verb as "to drench" means "to wet thoroughly" and this is in contradiction with rest of the sentence (e. "having lesser rain").

   Changed "drenches" to "shines".

10. Line 184, Replace "augmentation" with development.

    Done.

11. Line 259, can probably remove "site-level" as in situ literally means on site.

    Done.

12. Line 294, can replace "a regional and a time evolution" with "both a spatial and temporal view"

    Done.

13. Line 296, remove "s" from observations-based

Done.

14. Line 296, put name of products in bracket after mentioning there are two.

Done.

15. Line 296, reword end of sentence for clarity g. "ecosystem productivity in the GEOS simulations"

Done.

16. Line 297, move "Through upscaling using machine learning methods (Jung et al., 2020)" to the end of the

Done.

17. Line 312-31 It's a nudged run basically isn't it?

Yes, it is.

18. Line 321, maybe I missed it in the model description (section 2), are aerosols externally mixed in GEOS-GOCART?

All aerosol components are externally mixed in GEOS-GOCART.

19. Line 330 to 344, I hope I got this correctly, in nobbaer, R2 is only used by the physiology part of Catchment-CN, is that correct? Meaning that the rest of the land surface energy budget is calculated assuming R1 as in allaer.

Yes, that's correct.

20. Fig 1b is the same as Fig 1a. State in the legend what the error bars on 1c represent.

Sorry for this oversight. The correct 1b has been inserted. The legend now includes "The error bars on 1c indicate one standard deviation of the data within each 1km vertical layer."

21. Legend for Fig 2a and 2c specify the wavelength used for those AOD.

Done. These AODs are at 550nm.

22. Line 407, it would help the reader if the box defined here was depicted on some (all) of the contour Are all spatial averages quoted in the manuscript calculated over the same area?

Yes. We added the corresponding shaded area in Figure 2d and indicated it in line 421.

23. Line 421, maybe a reference here would be useful. Is vegetation not dark enough for MODIS dark target algorithm to perform well?

The statement here is given based on the personal study experience of our co-author. To be more rigorous, we delete this sentence.

24. Line 480 to 492. To avoid confusion, it would be better have full indexing for the radiative quantities, g. Rtot@toa instead of Rtop, Rdiff@srf instead of Rdiff …

Done.

25. Line 480 to 492, As the notation has changed from the rest of the manuscript, I believe the radiation quantities here are integrated over the full SW spectrum not just PAR, is that correct? Please clarify in text.

No. All Rs are over 400-700 nm as stated in the legend of Figure 9. We now also added this information in lines 538.

26. Line 480 to 492, how were the direct and diffused components calculated here? Was any delta-rescaling of the aerosol optical properties still applied?

Thanks for the question. We have added the following information in lines 524-529: "FastJX solves the 8-stream multiple scattering in atmospheric solar radiation transfer for direct and diffuse beams, using the exact scattering phase function and optical depths of atmospheric molecules, aerosols, and clouds, and provides photolytic intensities accurate typically to better than 3%, with worst case errors of no more 10% over a wide range of atmospheric conditions (Wild et al., 2000). No special approximations are needed to treat strongly forward-peaked phase functions." Optimizations were developed for the treatment of stratospheric $O_2$ and $O_3$ absorption (Bian et al., 2002) to speed up the simulation.

Following Chandrasekhar, these numerical solutions generally rely on expansion of the scattering phase function in Legendre polynomials. For large aerosols and cloud droplets, however, the scattering phase function is strongly forward

peaked and not easily represented by a truncated Legendre expansion (e.g., van de Hulst, 1981). The FastJX algorithm is based on a Legendre expansion of the exact scattering phase function, and thus no adjustment to optical depth or extinction coefficient is needed. A series of numerical tests, including expansion of forward-peaked phase functions to 160 terms, demonstrates that truncation of the Legendre expansion at 8 terms gives an accurate calculation of the mean specific intensity of the radiation field. The photolytic intensity (sometimes called 'actinic flux') is the sum of the direct solar flux plus the diffuse solar flux of 4ˇ-steradian integrated mean specific intensity.

Since FastJX solves the 8-stream multiple scattering problem, it does not apply a delta-scaling approximation, which is typically used to improve the accuracy of the two-stream approach to correct the calculated irradiances in strongly forward scattering conditions (Joseph et al., 1976; Kylling et al., 1995).

Joseph, J. H., Wiscombe, W. J., and Weinman, J. A., 1976: The delta-Eddington approximation for radiative flux transfer, J. Atmos. Sci. 33, 2452–2459.

Kylling, A., Stamnes, K., and Tsay, S. C., 1995: A reliable and efficient two-stream algorithm for spherical radiative transfer: Documentation of accuracy in realistic layered media, J. Atmos. Chem. 21, 115–150.

van de Hulst, H. C., 1981: Light Scattering by Small Particles, Dover, New York, p. 471.

27. Fig 7 legend, Should the unit for GPP be kg/m2/s?

The unit of GPP has been changed to be µg/m2/s.

28. Fig 7, A fifth timeseries representing the fractional / absolute change in GPP could be useful here.

Done.

29. Fig 7, Although the GPP for the site at 54W 15S is relatively high in GEOS (Fig 5C), it does not seem to be a very productive pixel in both FluxSat and FluxCom (Fig 5a and 5b) which makes sense as this is probably in the arc of deforestation. Is the tile in the model mostly covered by tall canopy PFTs or is dominated by lower grass?

Yes, this tile is located in the arc of deforestation and is covered by broadleaf deciduous temperate shrub (23%) and crop (77%).

30. Line 630, can be more specific and replace "quantities" by "GPP".

Done.

31. Fig 10 legend, BBAOD is labelled as the brown carbon part of the biomass burning aerosol, is that correct?

Corrected to "biomass burning aerosol".

32. Fig 10 legend, remove "for the ecosystem". Replace "dot-lines" by "dashed lines".

Done.

33. Fig 10 legend, replace "occurrence frequency" by "frequency of occurrence".

Done.

34. Fig 10 legend, "dGPP is 119.5% (201008) and 92.6%", hard to tell without seeing it plotted but from a quick and dirty "visual" interpolation, these numbers seems high. Can you confirm them

Yes. These numbers are high. We set a narrower range of values in plotting for easy visualization.

35. Line 628-29, sentence is already in Fig 10 legend.

The sentence has been deleted.

36. Line 635, Isn't this strong correlation to be expected from the experiment design? The atmospheric state should be pretty similar in allaer and nobbaer. The main ESM feedback allowed in nobbaer is from the vegetation that has received R2 instead of R1 isn't it? (see general comment #7 and specific comment #19). Anyway, such feedback probably has a negligeable impact on cloud fields (see g. Pedruzo-Bagazgoitia et al. 2017).

The atmospheric state including cloud fields is pretty similar in allaer and nobbaer (also see the answer to comment #6). However, the background atmospheric cloud fields fluctuate naturally each year. What we examined here is whether the effect of cloud interannual change overwhelms that of aerosol diffuse radiation fertilization effect. Thanks for the reference. It has been added in line 351-352.

37. Line 632-643, In this discussion, it would be useful to quote the GPP changes (both relative and absolute) averaged over the analysis period, maybe excluding 2010 as it is an unusual year. Additionally, these period mean quantities for GPP, DFPAR, DRPAR, AOD, CLDFRC could be incorporated in a table in the main manuscript (table S1/2 a,b,c,d are useful but clearly too detailed for the main manuscript).

Done. Table 3 has been added with the information of burning season multi-year (2011-2016) averaged GPP over the Amazon region from all four simulations. Also given in the table are the GPP, DFPAR, DRPAR, CLDFRC, and BBAOD averaged during Aug-Sept, 2011-2016 over the Amazon region in all-sky condition.

38. Line 647-652, This sentence is confusing. If cloudiness is similar in 2011 and 2010, how can you conclude from this that its effects are of second order compared to aerosol effects. Surely it won't have an impact if it doesn't change.

The sentence has been removed.

39. Line 653, a followed up to comment #36. This conclusion is based on the simulation results. Other environmental controls might have affected the vegetation productivity in the real world. Could you further support your conclusion by using the observational proxies (i.e. FluxCOM and FluxSat)? One way could be to compute a GPP anomaly as GPP for 2010 minus the mean of GPPs for the other years (excluding 2010) for FluxCOM, FluxSat and the allaer simulation. Then repeat the same calculation for 2011. Seven years might be a bit short for getting a representative mean, but I believe these two datasets have records longer than the analysed period.

This is a good suggestion and a good topic for future work. Such observational anomalous GPP (e.g., FluxCom) would contain all potential impacts from environment including interannual change in temperature, cloud, precipitation, etc. These other responses could be comparable or even larger than the biomass burning aerosol diffuse radiation fertilization effect that we studied here. For example, even the climate adjustments in response to the aerosol forcing can increase the efficiency of biochemical processes (67 to 100 TgCyr-1) comparably to the stimulation of diffuse light (65 to 110 TgCyr-1) over the Amazon region (Malavelle et al., 2019).

40. Line 657 to 664, This is the part of the paper where the most novel results are introduced. This paragraph could benefit from having some further explanation on the physical meaning of the anomalies calculated here. I sort of understand

ddX/dAOD as something similar to a radiative forcing (i.e. a difference in net radiative fluxes between a pair of simulations). Maybe it could be called susceptibility of the diffuse fertilization effect to BB aerosols. The double "dd" notation is a bit confusing and could be improved. It would be useful to use the same notation for the denominator (AOD) although I appreciate that the background aerosols in allaer and callaer cancel each other.

Thank you for pointing out this physical meaning. This certainly helps readers to understand our study. We have added the following sentences in lines 723-728. "This is, in principle, similar to the method of aerosol radiative forcing (RF) estimation ((i.e., estimating aerosol radiative effect (RE) with and without aerosol for present-day (pair1) and pre-industrial (pair2) and then deriving RF as a difference between the two pair REs). Here we study the sensitivity of the aerosol diffuse radiation fertilization effect to a unit change of AOD. We call it susceptibility of the diffuse fertilization effect to BB aerosols." Also BBAOD has been changed to AOD.

41. Fig 11, Instead of (or in addition to) the mean values for 2013/15, it could be useful to have a +/- one standard deviation marks around each means at each bin points. I expect much more variability in ddGPP/dAOD at low CF than at high CF. If so, there might be a larger range of CF where the aerosol effect could be detrimental to plant growth.

Vertical bars for one standard deviation have been added. The standard deviation is indeed larger at low CF.

**Additional reference:**

Pedruzo-Bagazgoitia, X., Ouwersloot, H. G., Sikma, M., vanHeerwaarden, C. C., Jacobs, C. M., and Vilà-Guerau deArellano, J.: Direct and Diffuse Radiation in the Shal-low Cumulus–Vegetation System: Enhanced and Decreased Evapotranspiration Regimes, J. Hydrometeorol., 18, 1731–1748, https://doi.org/10.1175/JHM-D-16-0279.1, 2017.

---

## Author Response (AR1)

We thank the reviewer for his/her comments. Please see our point-to-point reply below in blue.

**General comments**:

This study uses the NASA GEOS Earth System Model framework to investigate the impact of biomass burning aerosols and cloud cover on the Amazon region ecosystem productivity. This is a very interesting topic and the paper is clearly structured and generally well written.

However, while this work could certainly bring an important contribution to existing published studies on this topic, I think in its current form it still needs major revisions. I really hope this is something the authors can and will address in a revised manuscript.

**Major comments**:

1. A key question that needs to be addressed is whether the simulated response of GPP to changes in diffuse radiation fraction is realistic. More specifically, does the model accurately simulate observed GPP response to changes in diffuse, direct, and total surface radiation? And how does this simulated GPP response compare with other existing model estimates?

Answer: We strengthened the model evaluation with new Figures 5-6, 8, and S7-8 and discussions in lines 481-488. Following the evaluation approach in Malavelle et al. (2019), we evaluate our model's ability to simulate GPP on the global scale against FluxCom and FluxSat. As mentioned in section 2.2, FluxCom GPP is derived from surface measurements of carbon fluxes whereas FluxSat GPP is derived from satellite data. The comparison of global distribution of multiyear average GPP (Figure 5) and zonal mean multiyear average GPP (Figure 6) show that GEOS captures the GPP global distribution seen in the observations, with a GPP peak in tropics. The model does show a second peak in middle latitudes of the Southern Hemisphere but misses the observed peak in the Northern Hemisphere subtropics.

Malavelle et al., (2019) also conducted a similar evaluation for NPP. However, the MODIS NPP yearly data is currently unavailable due to unexpected errors in the input data related to persistent cloud cover (https://lpdaac.usgs.gov/products/mod17a3hv006/).

[Figure]

**Figure 5. 2010-2015 multiyear average GPP from FluxCom, FluxSat, and GEOS.**

[Figure]

**Figure 6. zonal mean of multiyear (2010-2015) average GPP from FluxCom, FluxSat, and GEOS.**

The regional multiyear GPP comparison among the three datasets was given and discussed already in the original submission (Figure 7). Although the above evaluations of global and regional multiyear average GPP in Figures 5-7 are needed for the examination of the model's fundamental mechanisms including photosynthesis, a more direct evaluation to address the model's accuracy in simulating observed GPP response to changes in diffuse and direct surface radiation is also needed and shown in Figure 8. Following the evaluation approach of Rap et al., (2015), we compared the GPP response to direct and diffuse light at two Amazon sites, Tapajos and Guyaflux. The figure clearly demonstrates that in the model, as in observations, diffuse light is more efficient in stimulating GPP (see lines 512-518).

[Figure]

**Figure 8. Observed (black) and GEOS modeled (blue) light response of GPP to direct (triangles) and diffuse (squares) photosynthetically active radiation (PAR) averaged over bins of 200 μmol quanta $m^2$ $s^1$ at (a) Tapajos and (b) Guyaflux. Error bars show 1 standard deviation of all values within a bin. The**

observation data are cited from Figure 2 of Rap et al. (2015) and the data period is 2002-2005 for Tapajos and 2006-2007 for Guyaflux, while model period is 2010-2016 for both sites.

To answer how this simulated GPP response compare with other existing model estimates, we summarized all relevant studies in Table 2 and added discussions correspondingly in lines 635-645.

**Table 2: Summary of model estimation of GPP increase in response to biomass burning aerosol over Amazon Basin**

| Study | This work | Malavelle2019 | Moreira2017 | Rap2015 | Strada2016 |
|---|---|---|---|---|---|
| GPP | 1.0% (dir+dif) | | 27% (dir+dif) | 0.7% (dir+dif) | 3.4% (dir+dif+clm)) |
| NPP | 1.5% (dir+dif) | 1.9 to 2.7% (dif+dir+clm) 1.5 to 2.6% (dif) -1.2 to -2.5% (dir) 1.6 to 2.4% (clm) | 52% (dir+dif) | 1.4% (dir+dif) | |
| Period | Annual average over 2010-2016 | Annual average over 30 model years, 2000 climate, | Sept., 2010 under cloud-free condition | Annual average over 1998-2007 | Annual average over 30 model years, 2000 climate |
| Atmospheric Model | GEOS ESM | HadGEM2-ES | BRAMS | | ModelE2 ESM |
| Running mode | replay | freeGCM | Regional model with ICBC from NCEP | offline | freeGCM |
| Vegetation model | Catchment-CN (using LSM4 for photosynthesis) | JULES | JULES | JULES | YiBs |
| Radiation model | RRTMG_SW | SOCRATES | CARMA | A two-stream radiative transfer model (Edwards and Slingo, 1996) | k-distribution approach with various updates (Schmidt et al., 2014) |
| Cloud model | Cloud microphysics model (Barahona et al., 2014) | | | Monthly mean clouds from ISCCP-D2 | a mass flux cumulus parameterization (Del Genio and Yao, 1993) |
| Aerosol model | GOCART | CLASSIC | CCATT | GLOMAP | OMA |
| BB emission | GFED4s | GFEDv2 1997-2006 average | 3BEM | GFED3 | IPCC AR5 |

dir, dif, and clm represent for direct radiation, diffuse radiation, and climate adjustment, respectively
3BEM: the Brazilian Biomass Burning Emission
BRAMS: Brazilian developments on the Regional Atmospheric Modeling System
CARMA: the Com-munity Aerosol and Radiation Model for Atmospheres
CCATT: a Eulerian transport model suitable to simulate trace gases and aerosols
CLASSIC: the Coupled Large-scale Aerosol Simulator for Studies In Climate
GLOMAP: The 3-D GLObal Model of Aerosol Processes Model
HadGEM2-ES:  The Hadley Centre Global Environment *Model*, version 2-Earth System
IPCC AR5: The Intergovernmental Panel on Climate Change Fifth Assessment Report
ISCCP-D2: the International Satellite Cloud Climatology Project
JULES: the Joint UK Land Environment Sim-ulator v3.0
OMA: One-Moment Aerosol,
SOCRATES: Suite Of Community RAdiative Transfer codes based on Edwards and Slingo
YIBs: The Yale Interactive Terrestrial Biosphere model

2. Why is the role of other climatic feedbacks associated with biomass burning aerosol emissions (e.g. reduction in leaf temperature) completely ignored, despite the fact that an ESM is used? While, the authors do acknowledge at the end of the paper (lines 716-719) that the aerosol induced changes in meteorological fields can also affect plant

growth, this seems to be a huge missed opportunity here. Malavelle et al. (2019) showed that the overall impact of biomass burning aerosols on NPP is the net result of multiple competing effects and it would be interesting to see if similar responses are simulated with the NASA GEOS ESM system.

As the reviewer pointed out, aerosols impact the ecosystem via various pathways: 1) adjusting radiation fluxes into the ecosystem, 2) changing atmospheric environment via its direct radiation effect, and 3) changing atmospheric environment via its semi- and indirect effect on cloud. In addition, biomass burning emission results not only in an increase of atmospheric aerosols, but also in the change of chemical gas components such as $CO_2$ and $O_3$. These gas tracers also have direct and indirect impact on the ecosystem. We would like to investigate these impacts in an incremental approach. In this study, we not only examine specifically the ecosystem response to the change of into-ecosystem radiation flux owing to biomass burning aerosols using NASA GEOS ESM, but also try to explain such impact from fundamental mechanism. We also investigate the role of the Amazon background cloud fields in tempering such impact. The importance of the latter study is explained in our following response to the C#3a-d of reviewer 2. We conducted our study using Replay mode so that we can exclude the compounding influence of aerosol-climate adjustments on atmospheric fields as we explained in text lines 309-317 and our response to reviewer 1's C#6 and reviewer 2's C#3d. To study the impact of aerosol-climate adjustments in the future, we need to switch the simulation configuration to freeGCM mode to let the model forecast meteorological fields by its governing equations all through its simulation period.

3. The second research objective (and the way it is addressed) is a bit unclear and it should be formulated and addressed much more clearly.

- 3a. It is evident that clouds have a substantial impact on the efficiency of the aerosol diffuse radiation effect, as they have a strong effect on diffuse radiation fraction. In a similar way it can be said that the aerosols have an impact on the efficiency of the diffuse radiation effect caused by clouds. So this in itself is not necessarily a research question.

  There are two distinctive features in clouds and aerosols that require us to treat them differently in their impact on the radiation flux to the ecosystem. First, like our distinction of natural and anthropogenic aerosols in their impact on air quality and climate, the cloud is a more natural phenomenon, while biomass burning aerosols (BBaer) can be, at least partially, controlled by humans. Second, clouds are much more efficient in controlling both direct and diffuse radiation fields than aerosols, see modified Figure 9 that added a thin-cloud mechanism. Based on the stronger efficiency of clouds in adjusting

radiation fields, it is worthwhile to investigate whether the same amount of BBaer could result in a very different impact on radiation fields under different background cloud conditions. As shown in Figure 14, under extreme environments, a unit increase of biomass burning AOD results in GPP increase by ~30 $kgm^{-2}s^{-1}$ in clear sky while GPP decrease by ~6 $kgm^{-2}s^{-1}$ in all cloud condition. Of course, the atmosphere of Amazon burning season could never be cloud free or completely clouded all the time in the real world. What is the potential range of the variation of Amazon clouds in burning seasons when the Amazon experiences environments of La Niña, normal years, and El Niño? To what extent does this range of cloud variation adjust the efficiency of "diffuse radiation fertilization effect" under the same emission strategy? These questions were not addressed clearly in previous studies, and we have tried to answer these questions in this study. We have added above discussion in lines 675-685.

- 3b. Is there a difference in the model between the simulated GPP response to changes in diffuse radiation fraction caused by aerosol changes and those caused by cloud cover changes?

  No. The GPP response to direct and diffuse radiations are calculated by integrating both aerosol and cloud fields. The point of the second research objective is that background cloud amount can adjust the impact of released biomass burning aerosol on direct and diffuse radiation, and consequently the biomass burning aerosol radiative diffusion fertilize effect.

- 3c. The fact that during the investigated period (lines 649-654), the interannual variation in regional cloudiness is small and therefore plays only a secondary role on the diffuse radiation fertilisation effect (compared to the dominant role played by the variation in biomass burning aerosol) is not surprising and does not really address the second research objective.

  The atmospheric radiation transfer theory tells us clouds dominate the atmospheric direct and diffuse radiation fields. It is worthwhile to investigate whether the emitted biomass burning emission could have a similar impact on surface radiation (and thereby similar impact on the ecosystem) under different atmospheric background environments. How sensitive is BBaer diffuse radiative fertilization effect to the different Amazon atmospheric conditions and to the potential bias in model cloud simulation? This is useful information for policy makers in controlling biomass burning emission.

Figure 14 indicates that dGPP can vary from 18.5 to 15.5 (kgm-2s-1) with a unit AOD of burning particles released to the atmosphere under the range of Amazon interannual cloud variation in dry season, which is 0.35 to 0.44 in our study period. In other words, there is ~20% dGPP uncertainty adjusted by background Amazon cloud in our studied period. Our work demonstrates quantitively the role of clouds in tempering aerosol diffuse radiation fertilization effect. We added the above discussion in lines 747-752.

- 3d. Lines 657-682: The cause for the difference between the 2013 and 2015 lines in Figure 11 is suggested to be the difference in cloud cover between the two years. I wonder whether this is indeed the case, since the results illustrated in Figure 11 are in fact for binned cloud fractions anyway? I would speculate they are in fact caused by the difference in (i) biomass burning emissions (they do matter in your calculated ddX/dBBAOD, which is defined in terms of both Pair1 and Pair2) and (ii) temperature and precipitation. This needs to be investigated and clarified.

First, following the suggestion of reviewer1 (S#41), we replotted this Figure by adding a ± one standard deviation representing potential variation. We can see much more variability in ddGPP/dAOD at low cloud cover (CF) than at high CF. The number of the occurrence frequency (%) in bins shown in the figure indicates that 2015 has more chances falling to low value CLDFRC bins than 2013.

Second, the impact from the potential variation of meteorologic fields in the four sensitivity runs is small. As we show in Figure S3 &4, the important relevant meteorologic fields of cloud fraction, surface T, and soil moisture over the studied area among the four simulations are very close. This phenomenon stems from the model feature we adopted in this study. In order to focus strictly on the ecosystem response to the into-ecosystem light perturbed by biomass burning aerosols, we use a Replay mode configuration in simulations. In Replay mode, every six hours, the model dynamic state (winds, pressure, temperature, and humidity) is set to the balanced state provided by the Modern-Era Retrospective analysis for Research and Applications, Version 2 (MERRA2) and then a six-hour forecast is performed until the next analysis is available. MERRA2 is the assimilation system that enables assimilation of modern hyperspectral radiance and microwave observations, ozone profiles observations, along with GPS-Radio Occultation datasets. Simulation with Replay is important in this study for two advantages: 1. Nudges GEOS dynamic fields to MERRA2 reanalysis ensuring  atmospheric conditions of the four simulations are close to each other, therefore, resulting

in more focus on the study of into-ecosystem radiative impact (also see answer to C#7 of reviewer 1), 2. Ensures the observational-constrained meteorological fields used in our study.

It is worth pointing out that previous studies listed in Table 2 were performed in a free running General Circulation Model (freeGCM) mode. In freeGCM mode, the model forecasts meteorological fields by its governing equations throughout its simulation period. Its dynamic system is self-consistent, which lets it be an ideal mode to be used for the study of aerosol-climate feedback. However, the meteorologic fields simulated with this approach may drift away for a long simulation period with a small uncertainty in initial conditions.

**Specific comments**:

1. Terminology: Why is the term "aerosol light fertilizer effect" being used instead of other already established terminology, e.g. diffuse radiation fertilization effect, Mercado et al (2009). I suggest the use of existing terminology to better integrate the work with other studies, but if the authors feel strongly about introducing this new terminology, a clear rationale for this should be provided.

   Our original thought focused more on the GPP response to the net effect of the radiation perturbed by biomass burning aerosol. We have changed the terminology to "diffuse radiation fertilization effect" to be consistent with previous studies such as Rap et al., (2015) and Mercado et al., (2019).

2. Why was the this particular period (i.e. 2010-2016) chosen? Can this be extended?

   Please refer to our answer to C#1 of reviewer 1.

3. Lines 100-101 and 140-142: It is not quite true that Malavelle et al (2019) did not consider the effect of clouds altering the diffuse radiation fertilisation effect. They do in fact discuss this and mention in their paper that "despite cloudiness affecting how much aerosols can interact with radiation, we notice that NPP is enhanced in the central part of the Amazon when BBA emissions are increased (Fig. 5)." So this needs to be reformulated and clarified in this paper to avoid confusions. This points also relates to my major comment 3, i.e. the need to better define and address the second research objective.

   Malavelle et al., (2019) gave a general direction that clouds affect the aerosol-radiation interaction. As we pointed out in our response to C#3a &3c, since

clouds typically dominate atmospheric radiation fields, we need to know how sensitive BBaer diffuse radiative fertilization effect is to the potential interannual variation of Amazon burning season clouds. Could the same amount of BBaer result in a very different impact on radiation fields under different background cloud conditions? To what extent does uncertainty in model cloud simulation affect our conclusion of BBaer diffuse radiative fertilization effect? We carried out investigations to address these questions specifically. We have modified the sentence in lines 100-103 as "Their study takes into account the dynamic feedback of short lifetime cloud fields. However, the authors have not explicitly quantified the impact of Amazon background clouds and their interannual changes in tempering the aerosol diffuse radiation fertilization effect (DRFE)."

4. Lines 140-142: The authors seem to have missed other relevant studies on this topic, such as Strada and Unger (2016) and Unger et al. (2017). Results presented in this work should also be compared and integrated with those from these other studies.

Thanks for introducing these two works. We have cited them (line 789). The GPP response to overall aerosol influence over the Amazon by Strada and Unger (2016) has been summarized in Table 2 as well.

5. Lines 403-406: It would be good to investigate a bit more the cause of the difference in observed and simulated SSA in August at Alta Floresta. What about other periods and other sites?

Following the reviewer' suggestion, we extended the SSA comparison to the whole year. This is also consistent with the whole year comparison for aerosol and gas tracer between GoAmazon campaign and GEOS model (Figure 1) and AOD among AERONET, MODIS, and GEOS (Figures 2).

The seasonality of the model SSA generally follows the pattern shown in AERONET observation with ~5% higher in its annual mean value. Certainly, there is room for the model to improve its optical property simulation, particularly for biomass burning aerosols. We have an on-going NASA funded project that aims specifically to study the impact of biomass burning aerosols and their chemical aging on optical properties and radiative forcing. The model reported relatively high SSA (Figure 2b) and high OC (Figure 1a) during the first half of August compared to the observations. Due to a high heterogeneity in aerosol distribution, it is challenging to capture aerosol plume in its exact time and location. There was a surge of biomass burning

pollution surrounding this station at the time shown in the CO observation (Figure 1b). Traditionally, CO is a good tracer for biomass burning study and its spatial distribution is relatively homogenous due to its longer lifetime.

We have gone through the AERONET observational data. There are only two AERONET stations that reported aerosol AOD measurements within the Amazon area and the study period, see Figure R2. However, station Arica is located at the bend of South America's western coast, which is influenced mainly by marine aerosol and local anthropogenic pollution. Only station Alta_Floresta, which is located at the center of the Amazon basin and had large biomass burning pollution, provides meaningful information for this study.

[Figure]

**Figure R2: two AERONET sites that located in the Amazon region and have observations in the studied period.**

6. Lines 458-461: Only comparing averages over Aug-Oct 2010-2016 for simulated SW radiation and CERES measurements can potentially mask important differences. Please include an assessment and discussion of model vs measurements agreement for the full time series (e.g. 2010-2016 time series of monthly means).

   Yes. The model-CERES SW radiation comparison for 2010-2016 time series of monthly mean over the Amazon region is now shown in Figure S6 (lines 476-477). Apparently, the GEOS model has a good Rsfc simulation for the full time series in its seasonality under both clear and all sky conditions. The multiyear annual average of Rsfc is about 2.8% and 3.6% higher in the GEOS simulation in clear sky and all sky conditions, respectively.

[Figure]

**Figure S6. The model-CERES shortwave surface downward radiation (Rsfc, Wm⁻²) comparison for 2010-2016 monthly mean time series over the Amazon region. The values given in legend are the multiyear average Rsfcs from the model and observation for all-sky and clear-sky conditions.**

7. Lines 470-477: Similarly to the evaluation of SW radiation, the evaluation of simulated GPP should be investigated in more detail, i.e. time series rather than just averages.

[Figure]

**Figure S8. The GPP comparison for 2010-2015 monthly mean time series over the Amazon region among FluxCom, FluxSat, and GEOS. The multiyear average GPP (gCm⁻²d⁻¹) are given in legend.**

Comparison of GPP timeseries over the Amazon region among FluxCom, FluxSat, and GEOS has been added (Figure S8). The seasonality of the three datasets is similar, which is high in boreal winter season and low in boreal summer season. However, the month with minimum GPP is shifted 1 to 2 months among the datasets (e.g., ~June for FluxSat, ~July for FluxCom, and ~August for GEOS). GEOS multiyear annual average GPP is close to the value of FluxSat but higher than that of FluxCom. Although there are few of observational sites available in FLUXNET 2015 Tier 1, Joiner et al. (2018) (hereafter J18) evaluated FluxSat GPP performance around Amazonia using

monthly data at 0.05◦ resolution for the tropical BR-Sa3 site (Figure 18a in J18). The evaluation showed that the high GPP values for this site produced by FluxSat (Figures 16 & 17 in J18) were supported by the flux tower values. The above discussion has been added in lines 501-509.

8. Figure 6: I would suggest the add another line corresponding to total radiation (i.e. the sum of the blue and red lines). This should help the discussion and better illustrate the point.

    Done.

9. Lines 479-516: This simulated response of total and diffuse surface radiation to different aerosol concentrations and cloud conditions needs to be evaluated against some observations. This is a key process to get right for this study and is not currently addressed in the paper. This relates to my major comment 1.

    In this section, we discussed the principal theory of how surface direct and diffuse light fluxes respond to the presence of aerosols and clouds using the photolysis model Fast-JX (Wild et al., 2000; Bian and Prather, 2002). FastJX solves the 8-stream multiple scattering in atmospheric solar radiation transfer for direct and diffuse beams, using the exact scattering phase function and optical depths of atmospheric molecules, aerosols, and clouds, and provides photolytic intensities accurate typically to better than 3%, with worst case errors of no more 10% over a wide range of atmospheric conditions (Wild et al., 2000) (also see our response to S#26 of reviewer 1). The model has served as a main core module for simulating atmospheric actinic fluxes and photochemistry in several global (e.g. GEOSCCM, GEOS-Chem, GFDL-AM4, MetUM) and regional (e.g. CMAQ, WRF-Chem) models.

    The model has also been evaluated against various other models that participated in an international multi-model comparison for solar fluxes and photolysis calculation organized by SPARC CCMVal2 (PhotoChem-2008 in Chipperfield et al., 2010). The primary goal is to improve model performance due to better calibration against laboratory and atmospheric measurements since some of the photochem-2008 models (e.g. TUV and NIWA) had been involved in previous campaigns, such as IPMMI and POLARIS. Recently, the model has been evaluated against the measurements from actinic flux spectroradiometers on board the NASA DC-8 during the Atmospheric Tomography (ATom) mission (Hair et al., 2018), which provided an extensive set of statistics on how clouds alter photolysis rates. In the aforementioned

evaluations, the fast-JX model is among the models with good performance. We have added above discussion in lines 524-534.

Chipperfield, M., Kinnison, D., Bekki, S., Bian, H., Brühl, C., Canty, T., et al. (2010). Stratospheric Chemistry (Chapter 6). In V. Eyring, T. G. Shepherd, & D. W. Waugh (Eds.), SPARC Report on the Evaluation of Chemistry-Climate Models. WCRP-132, WMO/TD No. 1526, SPARC Report No. 5 (pp. 191-252). Toronto: SPARC.

Hall, S. R., Ullmann, K., Prather, M. J., Flynn, C. M., Murray, L. T., Fiore, A. M., Correa, G., Strode, S. A., Steenrod, S. D., Lamarque, J.-F., Guth, J., Josse, B., Flemming, J., Huijnen, V., Abraham, N. L., and Archibald, A. T.: Cloud impacts on photochemistry: building a climatology of photolysis rates from the Atmospheric Tomography mission, Atmos. Chem. Phys., 18, 16809–16828, https://doi.org/10.5194/acp-18-16809-2018, 2018.

10. Figure 8 and lines 580-586: An Amazon regional average GPP increase of +9.9% resulting from an increase in DFPAR of 10% is substantially larger than other existing estimates, e.g. Rap et al. (2015), Malavelle et al. (2019). However, the corresponding percentage change in NPP (lines 597-601) seems closer to estimates from other studies. It is important to investigate this further and include a discussion on why this is the case (e.g. to what extent the GPP change is driven by changes in respiration and NPP, respectively). This point also relates to my major comment 1, regarding the need to validate the simulated GPP response to changes in diffuse/total radiation against observations and/or other existing model estimates.

Figure 8 (now Figure 11) shows the results in August 2010 when biomass burning emission was the maximum during the whole study period (see Figure S2a and Table S1a). On the 7-year burning seasons, the increase of Amazonian GPP estimated by our study was by 2.6% via a 3.8% increase in diffuse PAR (DFPAR) despite a 5.4% decrease in direct PAR (DRPAR) as we stated in our abstract and conclusion sections. The 7-year annual averaged GPP is only increased by 0.99% (Table 2), which is much lower than the value in burning seasons. We have compared our results with those from other model works in Table 2.

11. Lines 605-611: The comparison with Rap et al. (2015) is incorrect and misleading. Firstly, it is incorrect because the 0.5-4.2% range of NPP change in this study is an interannual range, while the 1.4-2.8% range from Rap et al. (2015) is an uncertainty range for the 1998-2007 average due to biomass burning emissions

uncertainty. The actual interannual range from Rap et al. (2015) can be inferred from their Fig. 4 and Fig. S5. Secondly, it is misleading as the two periods are different (2010-2016 vs 1998-2007), so any comparison of interannual ranges should also include a discussion on the interannual variability in biomass burning emissions during 1998-2016.

> Thanks for pointing this out. The NPP should increase ~1.4% with 1BBA emissions over 1998-2007 in Rap et al. (2015). We have corrected the number in the revised text (line 641). We also indicated the different simulation periods in the two studies (see Table 2) and showed the potential variation of BBaer emission over the period of 1998-2007 and 2010-2016 in Figure S2a.

12. Lines 702-703: "The cloud fraction at which BB aerosol switches from stimulating to inhibiting plant growth occurs at ~0.8." I think this is a potentially confusing statement as it only applies to the biomass burning aerosol loadings recorded during the period investigated here. In reality, as both cloud cover and aerosol concentrations affect the diffuse radiation fraction, this threshold does also depend on the aerosol loading. A more useful threshold would be one defined in terms of diffuse radiation fraction.

> This conclusion is made under the Amazon region within our study period. To avoid any potential confusion, we changed the sentence (lines 773-775) to "Over the Amazon region within our study period, the cloud fraction at which a unit AOD switches from stimulating to inhibiting plant growth occurs at ~0.8."

**Technical corrections**:

1. Line 32: "call here" should be "called here".

   > Done.

2. Line 124: missing supporting citation for the 40% value.

   > Li, W., R. Fu, and R. E. Dickinson (2006), Rainfall and its seasonality over the Amazon in the 21st century as assessed by the coupled models for the IPCC AR4, J. Geophys. Res., 111, D02111, doi:10.1029/2005JD006355.

3. Line 354: typo "metrological"

   > Corrected.

4. Lines 625-631: Description of figure is best included in the figure caption, with manuscript text dedicated to discussion of results.

   These lines have been deleted.

5. Lines 641-643: Please reformulate to avoid using "presumably" which is a bit too vague. A more precise statement would read much better.

   Change "presumably" to be "simply".

6. Lines 666-673: Why is a different font used in this paragraph?

   Font has been changed.

7. Lines 701-702: "Curiously, BB aerosols stimulate plant growth under clear-sky conditions but suppress it under full cloudiness conditions". I suggest removing the word "curiously"? This is in fact to be expected.

   Done.

**References**:

Strada, S. and Unger, N.: Potential sensitivity of photosynthesis and isoprene emission to direct radiative effects of atmospheric aerosol pollution, Atmos. Chem. Phys., 16, 4213–4234, https://doi.org/10.5194/acp-16-4213-2016, 2016.

Unger, N., Yue, X., and Harper, K. L.: Aerosol climate change effects on land ecosystem services, Faraday Discuss., 200, 121–142, https://doi.org/10.1039/c7fd00033b, 2017.